# FairGrad: Fairness Aware Gradient Descent

## Abstract

We address the problem of group fairness in classification, where the objective is to learn models that do not unjustly discriminate against subgroups of the population. Most existing approaches are limited to simple binary tasks or involve difficult to implement training mechanisms. This reduces their practical applicability. In this paper, we propose FairGrad, a method to enforce fairness based on a reweighting scheme that iteratively learns group specific weights based on whether they are advantaged or not. FairGrad is easy to implement and can accommodate various standard fairness definitions. Furthermore, we show that it is competitive with standard baselines over various datasets including ones used in natural language processing and computer vision.

## 1 Introduction

Fair Machine Learning addresses the problem of learning models that are free of any discriminatory behavior against a subset of the population. For instance, consider a company that develops a model to predict whether a person would be a suitable hire based on their biography. A possible source of discrimination here can be if, in the data available to the company, individuals that are part of a subgroup formed based on their gender, ethnicity, or other sensitive attributes, are consistently labelled as unsuitable hires regardless of their true competency due to historical bias. This kind of discrimination can be measured by a fairness notion called Demographic Parity (Calders et al., 2009). If the data is unbiased, another source of discriminate may stem from the model itself that consistently mislabel the competent individuals of a subgroup as unsuitable hires. This can be measured by a fairness notion called Equality of Opportunity (Hardt et al., 2016).

Several such fairness notions have been proposed in the literature as different problems call for different measures. These notions can be divided into two major paradigms, namely (i) Individual Fairness (Dwork et al., 2012; Kusner et al., 2017) where the idea is to treat similar individuals similarly regardless of the sensitive group they belong to, and (ii) Group Fairness (Calders et al., 2009; Hardt et al., 2016; Zafar et al., 2017a; Denis et al., 2021) where the underlying idea is that different sensitive groups should not be disadvantaged compared to an overall reference population. In this paper, we focus on group fairness in the context of classification.

The existing approaches for group fairness in Machine Learning may be divided into three main paradigms. First, pre-processing methods aim at modifying a dataset to remove any intrinsic unfairness that may exist in the examples. The underlying idea is that a model learned on this modified data is more likely to be fair (Dwork et al., 2012; Kamiran & Calders, 2012; Zemel et al., 2013; Feldman et al., 2015; Calmon et al., 2017). Then, post-processing approaches modify the predictions of an accurate but unfair model so that it becomes fair (Kamiran et al., 2010; Hardt et al., 2016; Woodworth et al., 2017; Iosifidis et al., 2019; Chzhen et al., 2019). Finally, in-processing methods aim at learning a model that is fair and accurate in a single step (Calders & Verwer, 2010; Kamishima et al., 2012; Goh et al., 2016; Zafar et al., 2017a;b; Donini et al., 2018; Krasanakis et al., 2018; Agarwal et al., 2018; Wu et al., 2019; Cotter et al., 2019; Iosifidis & Ntoutsi, 2019; Jiang & Nachum, 2020; Lohaus et al., 2020; Roh et al., 2020; Ozdayi et al., 2021). In this paper, we propose a new in-processing approach based on a reweighting scheme that may also be used as a kind of post-processing approach by fine-tuning existing classifiers.

**Motivation.** In-processing approaches can be further divided into several sub-categories (Caton & Haas, 2020). Common amongst them are methods that relax the fairness constraints under consideration to simplify the learning process (Zafar et al., 2017a; Donini et al., 2018; Wu et al., 2019).

```python
# The library is available as a part of the supplementary material.
from fairgrad.torch import CrossEntropyLoss

# Same as PyTorch's loss with some additional meta data.
# A fairness rate of 0.01 is a good rule of thumb for standardized data.
criterion = CrossEntropyLoss(y_train, s_train,
fairness_measure, fairness_rate=0.01)

# The dataloader and model are defined and used in the standard way.
for x, y, s in data_loader:
  optimizer.zero_grad()
  loss = criterion(model(x), y, s)
  loss.backward()
  optimizer.step()
```

Figure 1: A standard training loop where the PyTorch's loss is replaced by FairGrad's loss.

Indeed, standard fairness notions are usually difficult to handle as they are often non-convex and non-differentiable. Unfortunately, these relaxations may be far from the actual fairness measures, leading to sub-optimal models (Lohaus et al., 2020). Similarly, several approaches address the fairness problem by designing specific algorithms and solvers. This is, for example, done by reducing the optimization procedure to a simpler problem (Agarwal et al., 2018), altering the underlying solver (Cotter et al., 2019), or using adversarial learning (Raff & Sylvester, 2018). However, these approaches are often difficult to adapt to existing systems as they may require special training procedures or changes in the model. They are also often limited in the range of problems to which they can be applied (binary classification, two sensitive groups, . . . ). Furthermore, they may come with several hyperparameters that need to be carefully tuned to obtain fair models. The complexity of the existing methods might hinder their deployment in practical settings. Hence, there is a need for simpler methods that are straightforward to integrate in existing training loops.

**Contributions.** In this paper, we present FairGrad, a general purpose approach to enforce fairness for gradient descent based methods. We propose to dynamically update the weights of the examples after each gradient descent update to precisely reflect the fairness level of the models obtained at each iteration and guide the optimization process in a relevant direction. Hence, the underlying idea is to use lower weights for examples from advantaged groups than those from disadvantaged groups. Our method is inspired by recent reweighting approaches that also propose to change the importance of each group while learning a model (Krasanakis et al., 2018; Iosifidis & Ntoutsi, 2019; Jiang & Nachum, 2020; Roh et al., 2020; Ozdayi et al., 2021). We discuss these works in Appendix A.

A key advantage of FairGrad is that it is straightforward to incorporate into standard gradient based solvers that support examples reweighing like Stochastic Gradient Descent. Hence, we developed a Python library (provided in the supplementary material) where we augmented standard PyTorch losses to accommodate our approach. From a practitioner point of view, it means that using FairGrad is as simple as replacing their existing loss from PyTorch with our custom loss and passing along some meta data, while the rest of the training loop remains identical. This is illustrated in Figure 1. It is interesting to note that FairGrad only brings one extra hyper-parameter, the fairness rate, besides the usual optimization ones (learning rates, batch size, . . . ).

Another advantage of Fairgrad is that, unlike the existing reweighing based approaches which often focus on specific settings, it is compatible with various group fairness notions, including exact and approximate fairness, can handle both multiple sensitive groups and multiclass problems, and can fine tune existing unfair models. Through extensive experiments, we show that, in addition to its versatility, FairGrad is competitive with several standard baselines in fairness on both standard datasets as well as complex natural language processing and computer vision tasks.

## 2 PROBLEM SETTING AND NOTATIONS

In the remainder of this paper, we assume that we have access to a feature space $\mathcal{X}$, a finite discrete label space $\mathcal{Y}$, and a set $\mathcal{S}$ of values for the sensitive attribute. We further assume that there exists an unknown distribution $\mathcal{D} \in \mathcal{D}_{\mathcal{Z}}$ where $\mathcal{D}_{\mathcal{Z}}$ is the set of all distributions over $\mathcal{Z} = \mathcal{X} \times \mathcal{Y} \times \mathcal{S}$ and

that we only get to observe a finite dataset $\mathcal{T} = \{(x_i, y_i, s_i)\}_{i=1}^n$ of $n$ examples drawn i.i.d. from $\mathcal{D}$. Our goal is then to learn an accurate model $h_\theta \in \mathcal{H}$, with learnable parameters $\theta \in \mathbb{R}^D$, such that $h_\theta : \mathcal{X} \to \mathcal{Y}$ is fair with respect to a given fairness definition that depends on the sensitive attribute. In Section 2.1, we formally define the fairness measures that are compatible with our approach and provide several examples of popular notions that are compatible with our method. Finally, for the ease of presentation, throughout this paper we slightly abuse the notation $\mathbb{P}(E)$ and use it to represent both the true probability of an event $E$ and its estimated probability from a finite sample.

## 2.1 Fairness Definition

We assume that the data may be partitioned into $K$ disjoint groups denoted $\mathcal{T}_1, \ldots, \mathcal{T}_k, \ldots, \mathcal{T}_K$ such that $\bigcup_{k=1}^K \mathcal{T}_k = \mathcal{T}$ and $\bigcap_{k=1}^K \mathcal{T}_k = \emptyset$. These groups highly depend on the fairness notion under consideration. They might correspond to the usual sensitive groups, as in Accuracy Parity (see Example 1), or might be subgroups of the usual sensitive groups, as in Equalized Odds (see Example 2 in the appendix). For each group, we assume that we have access to a function $F_k : \mathcal{D}_\mathcal{Z} \times \mathcal{H} \to \mathbb{R}$ such that $F_k > 0$ when the group $k$ is advantaged and $F_k < 0$ when the group $k$ is disadvantaged. Furthermore, we assume that the magnitude of $F_k$ represents the degree to which the group is (dis)advantaged. Finally, we assume that each $F_k$ can be rewritten as follows:

$$F_k(\mathcal{T}, h_\theta) = C_k^0 + \sum_{k'=1}^K C_k^{k'} \mathbb{P}(h_\theta(x) \neq y | \mathcal{T}_{k'}) \tag{1}$$

where the constants $C$ are group specific and independent of $h_\theta$. The probabilities $\mathbb{P}(h_\theta(x) \neq y | \mathcal{T}_{k'})$ represent the error rates of $h_\theta(x)$ over each group $\mathcal{T}_{k'}$ with a slight abuse of notation. Below, we show that Accuracy Parity (Zafar et al., 2017a) respects this definition. In Appendix B, we show that Equality of Opportunity (Hardt et al., 2016), Equalized Odds (Hardt et al., 2016), and Demographic Parity (Calders et al., 2009) also respect this definition.

**Example 1 (Accuracy Parity (AP) (Zafar et al., 2017a)).** A model $h_\theta$ is fair for Accuracy Parity when the probability of being correct is independent of the sensitive attribute, that is, $\forall r \in \mathcal{S}$

$$\mathbb{P}(h_\theta(x) = y \,|\, s = r) = \mathbb{P}(h_\theta(x) = y).$$

It means that we need to partition the space into $K = |\mathcal{S}|$ groups and, $\forall r \in \mathcal{S}$, we define $F_{(r)}$ as the fairness level of group $(r)$

$$F_{(r)}(\mathcal{T}, h_\theta) = \mathbb{P}(h_\theta(x) \neq y) - \mathbb{P}(h_\theta(x) \neq y \,|\, s = r)$$
$$= (\mathbb{P}(s = r) - 1)\mathbb{P}(h_\theta(x) \neq y \,|\, s = r) + \sum_{(r') \neq (r)} \mathbb{P}(s = r')\mathbb{P}(h_\theta(x) \neq y \,|\, s = r')$$

where the law of total probability was used to obtain the last equality. Thus Accuracy Parity satisfies all our assumptions with $C_{(r)}^{(r)} = \mathbb{P}(s = r) - 1$, $C_{(r)}^{(r')} = \mathbb{P}(s = r')$ with $r' \neq r$, and $C_{(r)}^0 = 0$.

## 3 FairGrad

In this section, we present FairGrad, the main contribution of this paper. We begin by discussing FairGrad for exact fairness and then present an extension to handle $\epsilon$-fairness.

## 3.1 FairGrad for Exact Fairness

To introduce our method, we first start with the following optimization problem that is standard in fair machine learning (Cotter et al., 2019)

$$\arg\min_{h_\theta \in \mathcal{H}} \mathbb{P}(h_\theta(x) \neq y)$$
$$\text{s.t. } \forall k \in [K], F_k(\mathcal{T}, h_\theta) = 0. \tag{2}$$

Then, using Lagrange multipliers, denoted $\lambda_1, \ldots, \lambda_K$, we obtain an unconstrained objective that should be minimized for $h_\theta \in \mathcal{H}$ and maximized for $\lambda_1, \ldots, \lambda_K \in \mathbb{R}$:

$$\mathcal{L}(h_\theta, \lambda_1, \ldots, \lambda_K) = \mathbb{P}(h_\theta(x) \neq y) + \sum_{k=1}^K \lambda_k F_k(\mathcal{T}, h_\theta). \tag{3}$$

To solve this problem, we propose to use an alternating approach where the hypothesis and the multipliers are updated one after the other[1].

**Updating the Multipliers.** To update $\lambda_1, \ldots, \lambda_K$, we will use a standard gradient ascent procedure. Hence, given that the gradient of Problem (3) is

$$
\nabla_{\lambda_1, \ldots, \lambda_K} \mathcal{L}\left(h_\theta, \lambda_1, \ldots, \lambda_K\right) = \begin{pmatrix} F_1(\mathcal{T}, h_\theta) \\ \vdots \\ F_K(\mathcal{T}, h_\theta) \end{pmatrix}
$$

we have the following update rule $\forall k \in [K]$:

$$
\lambda_k^{T+1} = \lambda_k^T + \eta_\lambda F_k\left(\mathcal{T}, h_\theta^T\right)
$$

where $\eta_\lambda$ is a rate that controls the importance of each update. In the experiments, we use a constant fairness rate of $0.01$ as our initial tests showed that it is a good rule of thumb when the data is properly standardized.

**Updating the Model.** To update the parameters $\theta \in \mathbb{R}^D$ of the model $h_\theta$, we use a standard gradient descent approach. However, first, we notice that given the fairness notions considered, Equation (3) can be rewritten as

$$
\mathcal{L}\left(h_\theta, \lambda_1, \ldots, \lambda_K\right) = \sum_{k=1}^K \mathbb{P}\left(h_\theta(x) \neq y | \mathcal{T}_k\right) \left[ \mathbb{P}\left(\mathcal{T}_k\right) + \sum_{k'=1}^K C_{k'}^k \lambda_{k'} \right] + \sum_{k=1}^K \lambda_k C_k^0. \tag{4}
$$

where $\sum_{k=1}^K \lambda_k C_k^0$ is independent of $h_\theta$ by definition. Hence, at iteration $t$, the update rule becomes

$$
\theta^{T+1} = \theta^T - \eta_\theta \sum_{k=1}^K \left[ \mathbb{P}\left(\mathcal{T}_k\right) + \sum_{k'=1}^K C_{k'}^k \lambda_{k'} \right] \nabla_\theta \mathbb{P}\left(h_\theta(x) \neq y | \mathcal{T}_k\right)
$$

where $\eta_\theta$ is the usual learning rate that controls the importance of each parameter update. Here, we obtain our group specific weights $\forall k, w_k = \left[ \mathbb{P}\left(\mathcal{T}_k\right) + \sum_{k'=1}^K C_{k'}^k \lambda_{k'} \right]$, that depend on the current fairness level of the model through $\lambda_1, \ldots, \lambda_K$, the relative size of each group through $\mathbb{P}\left(\mathcal{T}_k\right)$, and the fairness notion under consideration through the constants $C$. The exact values of these constants are given in Section 2.1 and Appendix B for various group fairness notions. Overall, they are such that, at each iteration, the weights of the advantaged groups are reduced and the weights of the disadvantaged groups are increased.

The main limitation of the above update rule is that one needs to compute the group-wise gradients $\nabla_\theta \mathbb{P}\left(h_\theta(x) \neq y | \mathcal{T}_k\right) = \frac{1}{n_k} \sum_{(x,y) \in \mathcal{T}_k} \nabla_\theta \mathbb{I}_{\{h_\theta(x) \neq y\}}$. Here, $\mathbb{I}_{\{h_\theta(x) \neq y\}}$ is the indicator function, also called the $0-1$-loss, that is $1$ when $h_\theta(x) \neq y$ and $0$ otherwise. Unfortunately, this usually does not provide meaningful optimization directions. To address this issue, we follow the usual trend in machine learning and replace the $0-1$-loss with one of its continuous and differentiable surrogates that provides meaningful gradients. For instance, in our experiments, we use the cross entropy loss.

## 3.2 COMPUTATIONAL OVERHEAD OF FAIRGRAD.

We summarize our approach in Algorithm 1. We consider batch gradient descent rather than full gradient descent as it is a popular optimization scheme. We empirically investigate the impact of the batch size in Section 4.7. We use italic font to highlight the steps inherent to FairGrad that do not appear in classic batch gradient descent. The main difference is Step 5, that is the computation of the fairness levels for each group. However, these can be cheaply obtained from the predictions of $h_\theta^{(t)}$ on the current batch which are always available since they are also needed to compute the gradient. Hence, the computational overhead of FairGrad is very limited.

---

[1]It is worth noting that, here, we do not have formal duality guarantees and that the problem is not even guaranteed to have a fair solution. Nevertheless, the approach seems to work well in practice as can be seen in the experiments.

---

**Algorithm 1** FairGrad for Exact Fairness

---

**Input**: Groups $\mathcal{T}_1, \ldots, \mathcal{T}_K$, Functions $F_1, \ldots, F_K$, Function class $\mathcal{H}$ of models $h_\theta$ with parameters $\theta \in \mathbb{R}^D$, Learning rates $\eta_\lambda, \eta_\theta$, and Iterator $iter$ that returns batches of examples.
**Output**: A fair model $h_\theta^*$.

1:  Initialize *the group specific weights* and the model.
2:  **for** B in $iter$ **do**
3:    Compute the predictions of the current model on the batch B.
4:    Compute the group-wise losses using the predictions.
5:    *Compute the current fairness level using the predictions and update the group-wise weights.*
6:    Compute the overall *weighted* loss using the *group-wise weights*.
7:    Compute the gradients based on the loss and update the model.
8:  **end for**
9:  **return** the trained model $h_\theta^*$

---

### 3.3 Importance of Negative Weights.

A key property of FairGrad is that we allow the use of negative weights throughout the optimization process, that is $\left[ \mathbb{P}\left(\mathcal{T}_k\right) + \sum_{k'=1}^{K} C_{k'}^k \lambda_{k'} \right]$ may become negative, while existing methods often restrict themselves to positive weights (Roh et al., 2020; Iosifidis & Ntoutsi, 2019; Jiang & Nachum, 2020). In this Section, we show that these negative weights are important as they are sometimes necessary to learn fair models. Hence, in the next lemma, we provide sufficient conditions so that negative weights are mandatory if one wants to enforce Accuracy Parity.

**Lemma 1** (Negative weights are necessary.). *Assume that the fairness notion under consideration is Accuracy Parity (see Example 1). Let $h_\theta^*$ be the most accurate and fair model. Then using negative weights is necessary as long as*

$$\min_{\substack{h_\theta \in \mathcal{H} \\ h_\theta \, unfair}} \max_{\mathcal{T}_k} \mathbb{P}\left(h_\theta(x) \neq y | \mathcal{T}_k\right) < \mathbb{P}\left(h_\theta^*(x) \neq y\right).$$

*Proof.* The proof is provided in Appendix C.  □

The previous condition can sometimes be verified in practice. As a motivating example, assume a binary setting with only two sensitive groups $\mathcal{T}_1$ and $\mathcal{T}_{-1}$. Let $h_\theta^{-1}$ be the model minimizing $\mathbb{P}\left(h_\theta(x) \neq y | \mathcal{T}_{-1}\right)$ and assume that $\mathbb{P}\left(h_\theta^{-1}(x) \neq y\right) < \mathbb{P}\left(h_\theta^{-1}(x) \neq y | \mathcal{T}_{-1}\right)$, that is group $\mathcal{T}_{-1}$ is disadvantaged for accuracy parity. Given $h_\theta^*$ the most accurate and fair model, we have

$$\min_{\substack{h_\theta \in \mathcal{H} \\ h_\theta \, unfair}} \max_{\mathcal{T}_k} \mathbb{P}\left(h_\theta(x) \neq y | \mathcal{T}_k\right) = \mathbb{P}\left(h_\theta^{-1}(x) \neq y | \mathcal{T}_{-1}\right) < \mathbb{P}\left(h_\theta^*(x) \neq y\right)$$

as otherwise we would have a contradiction since the fair model would also be the most accurate model for group $\mathcal{T}_{-1}$ since $\mathbb{P}\left(h_\theta^*(x) \neq y\right) = \mathbb{P}\left(h_\theta^*(x) \neq y | \mathcal{T}_{-1}\right)$ by definition of Accuracy Parity. In other words, a dataset where the most accurate model for a given group still disadvantages this group requires negative weights.

### 3.4 FairGrad for $\epsilon$-fairness

In the previous section, we mainly considered exact fairness and we showed that this could be achieved by using a reweighting approach. In fact, we can extend this procedure to the case of $\epsilon$-fairness where the fairness constraints are relaxed and a controlled amount of violations is allowed. Usually, $\epsilon$ is a user defined parameter but it can also be set by the law, as it is the case with the $80\%$ rule in the US. The main difference with the exact fairness case is that each equality constraint in Problem (2) is replaced with two inequalities of the form

$$\forall k \in [K], F_k(\mathcal{T}, h_\theta) \leq \epsilon$$
$$\forall k \in [K], F_k(\mathcal{T}, h_\theta) \geq -\epsilon.$$

The main consequence is that we need to maintain twice as many Lagrange multipliers and that the group-wise weights are slightly different. Since the two procedures are similar, we omit the details here but provide them in Appendix D for the sake of completeness.

# 4 EXPERIMENTS

In this section, we present several experiments that demonstrate the competitiveness of FairGrad as a procedure to learn fair models in a classification setting. We begin by presenting results over standard fairness datasets and a Natural language Processing dataset in Section 4.4. We then study the behaviour of the $\epsilon$-fairness variant of FairGrad in Section 4.5. Next, we showcase the fine-tuning ability of FairGrad on a Computer Vision dataset in Section 4.6. Finally, we investigate the impact of batch size on the learned model in Section 4.7.

## 4.1 DATASETS

In the main paper, we consider $4$ different datasets and postpone the results on another $6$ datasets to Appendix E as they follow similar trends. We also postpone the detailed descriptions of these datasets as well as the pre-processing steps.

On the one hand, we consider commonly used fairness datasets, namely **Adult Income** (Kohavi, 1996) and **CelebA** (Liu et al., 2015). Both are binary classification datasets with binary sensitive attributes (gender). We also consider a variant of the Adult Income dataset where we add a second binary sensitive attribute (race) to obtain a dataset with $4$ disjoint sensitive groups.

On the other hand, to showcase the wide applicability of FairGrad, we consider the **Twitter Sentiment**[2] (Blodgett et al., 2016) dataset from the Natural Language Processing community. It consists of $200k$ tweets with binary sensitive attribute (race) and binary sentiment score. We also employ the **UTKFace** dataset[3] (Zhang et al., 2017) from the Computer Vision community. It consists of $23,708$ images tagged with race, age, and gender.

## 4.2 PERFORMANCE MEASURES

For fairness, we consider the four measures introduced in Section 2.1 and Appendix B, namely Equalized Odds (EOdds), Equality of Opportunity (EOpp), Accuracy Parity (AP), and Demographic Parity (DP). For each specific fairness notion, we report the average absolute fairness level of the different groups over the test set, that is $\frac{1}{K} \sum_{k=1}^{K} |F_k(\mathcal{T}, h_\theta)|$ (lower is better). To assess the utility of the learned models, we use their accuracy levels over the test set, that is $\frac{1}{n} \sum_{i=1}^{n} \mathbb{I}_{h_\theta(x_i)=y_i}$ (higher is better). All the results reported are averaged over 5 independent runs and standard deviations are provided. Note that, in the main paper, we graphically report a subset of the results over the aforementioned datasets. We provide detailed results in Appendix E, including the missing pictures as well as complete tables with accuracy levels, fairness levels, and fairness level of the most well-off and worst-off groups for all the relevant methods.

## 4.3 METHODS

We compare FairGrad to $6$ different baselines, namely (i) Unconstrained, which is oblivious to any fairness measure and trained using a standard batch gradient descent method, (ii) an Adversarial mechanism (Goodfellow et al., 2014) using a gradient reversal layer (Ganin & Lempitsky, 2015), similar to GRAD-Pred (Raff & Sylvester, 2018), where an adversary, with an objective to predict the sensitive attribute, is added to the unconstrained model, (iii) BiFair (Ozdayi et al., 2021), (iv) FairBatch (Roh et al., 2020), (v) Constraints (Cotter et al., 2019), a non-convex constrained optimization method, and (vi) Weighted ERM where each example is reweighed based on the size of the sensitive group the example belongs to.

In all our experiments, we consider two different hypothesis classes. On the one hand, we use linear models implemented in the form of neural networks with no hidden layers. On the other hand, we use a more complex, non-linear architecture with three hidden layers of respective sizes $128$, $64$, and $32$. We use ReLU as our activation function with batch norm normalization and dropout set to $0.2$. In both cases, we optimize the cross-entropy loss. We provide the exact setup and hyper-parameter tuning details for all the methods in Appendix E.1.

---

[2]http://slanglab.cs.umass.edu/TwitterAAE/
[3]https://susanqq.github.io/UTKFace/

In several experiments, we only consider subsets of the baselines due to the limitations of the methods. For instance, BiFair was designed to handle binary labels and binary sensitive attributes and thus is not considered for the datasets with more than two sensitive groups or more than two labels. Furthermore, we implemented it using the authors code that is freely available online but does not include AP as a fairness measure, thus we do not report results related to this measure for BiFair. Similarly, we also implemented FairBatch from the authors code which does not support AP as a fairness measure, thus we also exclude it from the comparison for this measure. For Constraints, we based our implementation on the publicly available authors library but were only able to reliably handle linear models and thus we do not consider this baseline for non-linear models. Finally, for Adversarial, we used our custom made implementation. However, it is only applicable when learning non-linear models since it requires at least one hidden layer to propagate its reversed gradient.

## 4.4 RESULTS FOR EXACT FAIRNESS

We report the results over the Adult Income dataset using a linear model, the Adult Income dataset with multiple groups with a non-linear model, and the Twitter sentiment dataset using both linear and nonlinear models in Figures 2, 3, and 4 respectively. In these figures, the best methods are closer to the bottom right corner. If a method is closer to the bottom left corner, it has good fairness but reduced accuracy. Similarly, a method closer to the top right corner has good accuracy but poor fairness, that is it is close to the unconstrained model.

The main take-away from these experiments is that there is no fairness enforcing method that is consistently better than the others. All of them have strengths, that is datasets and fairness measures where they obtain good results, and weaknesses, that is datasets and fairness measures for which they are sub-optimal. For instance, FairGrad achieves better fairness levels for EOdds and EOpp over the Adult dataset with a linear model. However, it pays a price in terms of accuracy in those settings. Similarly, FairBatch induces better accuracy than the other approaches over Adult with linear model and EOdds and only pays a small price in terms of fairness. However, it is significantly worse in terms of fairness over the Adult Multigroup dataset with a non-linear model. Finally, BiFair is sub-optimal on Adult with EOpp, while being comparable to the other approaches on the Twitter Sentiment dataset. We observed similar trends on the other datasets, available in Appendix E.3, with different methods coming out on top for different datasets and fairness measures.

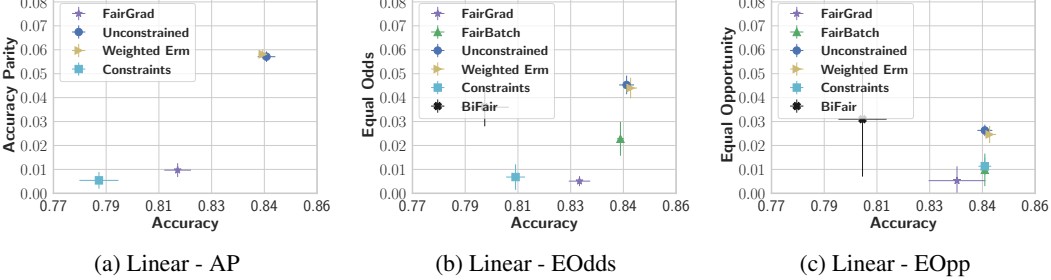

(a) Linear - AP        (b) Linear - EOdds        (c) Linear - EOpp

Figure 2: Results for the Adult dataset using Linear Models.

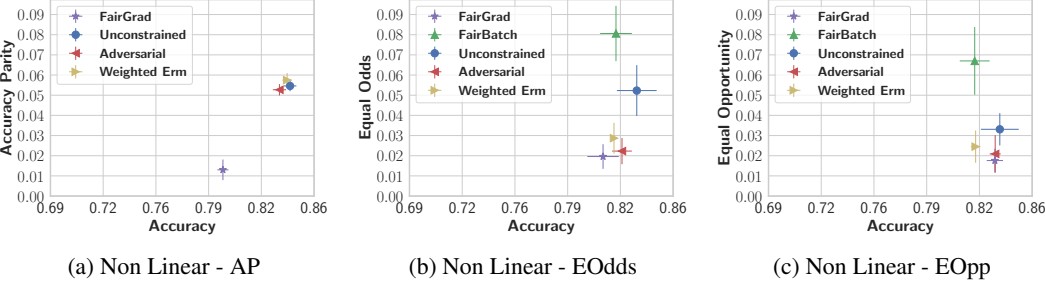

(a) Non Linear - AP        (b) Non Linear - EOdds        (c) Non Linear - EOpp

Figure 3: Results for the Adult Multigroup dataset using Non Linear models.

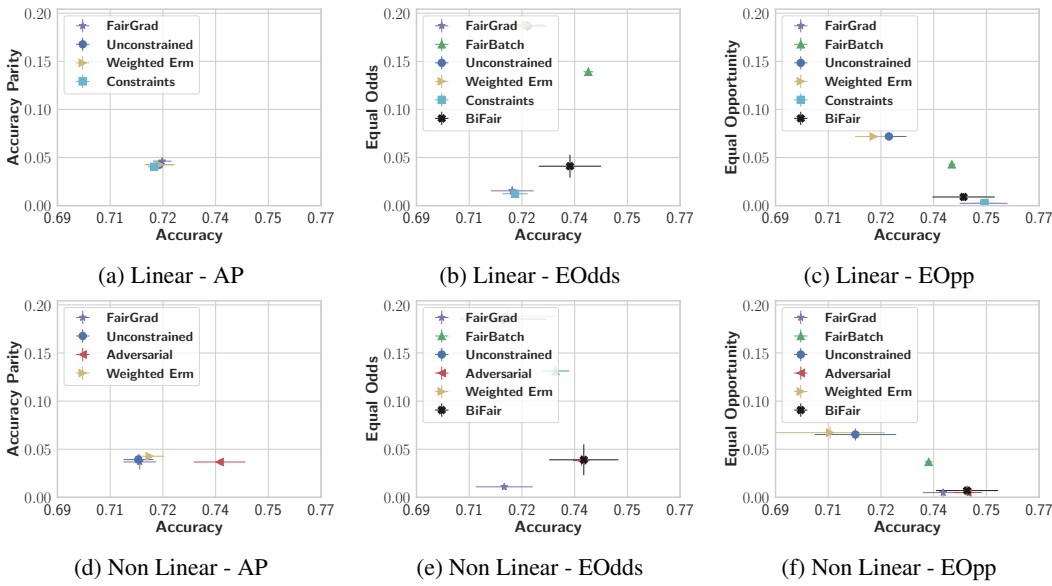

Figure 4: Results for the Twitter Sentiment dataset for Linear and Non Linear Models.

## 4.5 ACCURACY FAIRNESS TRADE-OFF

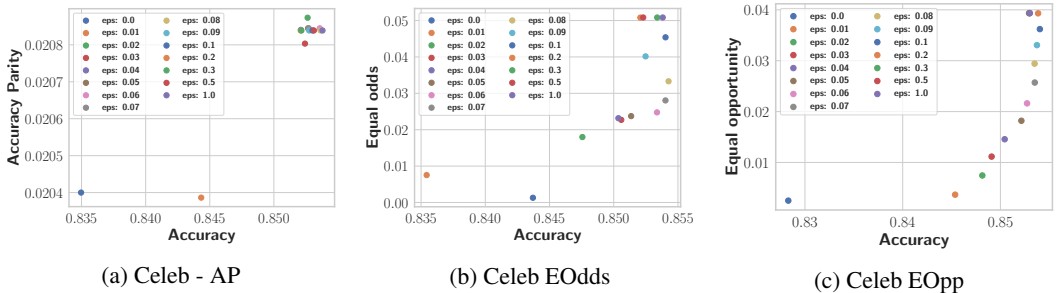

Figure 5: Results for CelebA with different fairness measure using Linear models. The Unconstrained Linear model achieves a test accuracy of $0.8532$ with fairness level of $0.0499$ for EOdds, $0.0204$ for AP, and $0.0387$ for EOpp.

In this second set of experiments, we demonstrate the capability of FairGrad to support approximate fairness (see Section 3.4). In Figure 5, we show the performances, as accuracy-fairness pairs, of several models learned on the CelebA dataset by varying the fairness level parameter $\epsilon$. These results suggest that FairGrad respects the constraints well. Indeed, the average absolute fairness level (across all the groups, see Section 4.2) achieved by FairGrad is either the same or less than the given threshold. It is worth mentioning that FairGrad is designed to enforce $\epsilon$-fairness for each constraint individually which is slightly different from the summarized quantity displayed here. Finally, as the fairness constraint is relaxed, the accuracy of the model increases, reaching the same performance as the Unconstrained classifier when the fairness level of the latter is below $\epsilon$.

## 4.6 FAIRGRAD AS A FINE-TUNING PROCEDURE

While FairGrad has primarily been designed to learn fair classifiers from scratch, it can also be used to fine-tune an existing classifier to achieve better fairness. To showcase this possibility, we fine-tune the ResNet18 (He et al., 2016) model, developed for image recognition, over the UTKFace dataset (Zhang et al., 2017), consisting of human face images tagged with Gender, Age, and Race information. Following the same process as Roh et al. (2020), we use Race as the sensitive attribute and consider two scenarios where either the gender (binary) with Demographic Parity as the fairness

Table 1: Results for the UTKFace dataset where a ResNet18 is fine-tuned using different strategies.

| Method | s=Race ; y=Gender | | s=Race ; y=Age | |
|---|---|---|---|---|
| | Accuracy | DP | Accuracy | EOdds |
| Unconstrained | $0.8691 \pm 0.0075$ | $0.0448 \pm 0.0066$ | $0.6874 \pm 0.0080$ | $0.0843 \pm 0.0089$ |
| FairGrad | $0.8397 \pm 0.0085$ | $0.0111 \pm 0.0064$ | $0.6491 \pm 0.0082$ | $0.0506 \pm 0.0059$ |

Table 2: Effect of the batch size on the CelebA dataset with Linear Models and EOdds as the fairness measure.

| Batch Size | 8 | 16 | 32 | 64 | 128 | 256 | 512 | 1024 | 2048 |
|---|---|---|---|---|---|---|---|---|---|
| Accuracy | 0.8186 | 0.8234 | 0.8215 | 0.8268 | 0.8273 | 0.8286 | 0.8292 | 0.8289 | 0.8303 |
| Accuracy Std | 0.0013 | 0.006 | 0.0028 | 0.0025 | 0.0031 | 0.0008 | 0.0027 | 0.0017 | 0.0031 |
| Fairness | 0.0031 | 0.0091 | 0.0045 | 0.0036 | 0.0051 | 0.0046 | 0.004 | 0.0038 | 0.0057 |
| Fairness Std | 0.0042 | 0.0062 | 0.0012 | 0.0014 | 0.0025 | 0.0032 | 0.0026 | 0.0019 | 0.0018 |

measure or age (multi-valued) with Equalized Odds as fairness measure are used as the target label. The results are displayed in Table 1. In both settings, FairGrad is able to learn models that are more fair than an Unconstrained fine-tuning procedure, albeit at the expense of accuracy.

### 4.7 IMPACT OF THE BATCH-SIZE

In this last set of experiment, we evaluate the impact of batch size on the fairness and accuracy level of the learned model. Indeed, at each iteration, in order to minimize the overhead associated with FairGrad (see Section 3.1), we update the weights using the fairness level of the model estimated solely on the current batch. When these batches are small, these estimates are unreliable and might lead the model astray. This can be observed in Table 2 where we present the performances of several linear models learned with different batch sizes on the CelebA dataset. On the one hand, for very small batch sizes, the learned models tends to have slightly lower accuracy and larger standard deviation in fairness levels. On the other hand, with a sufficiently large batch size, in this case $64$ and above, the learned models are close to be perfectly fair. Furthermore, they obtain reasonable levels of accuracy since the Unconstrained model has an accuracy of $0.8532$ for this problem.

## 5 CONCLUSION

In this paper, we proposed FairGrad, a fairness aware gradient descent approach based on a reweighting scheme. We showed that it can be used to learn fair models for various group fairness definitions and is able to handle multiclass problems as well as settings where there is multiple sensitive groups. We empirically showed the competitiveness of our approach against several baselines on standard fairness datasets and on a Natural Language Processing task. We also showed that it can be used to fine-tune an existing model on a Computer Vision task. Finally, since it is based on gradient descent and has a small overhead, we believe that FairGrad could be used for a wide range of applications, even beyond classification.

**Limitations and Societal Impact.** While appealing, FairGrad also has limitations. It implicitly assumes that a set of weights that would lead to a fair model exists but this might be difficult to verify in practice. Thus, even if in our experiments FairGrad seems to behave quite well, a practitioner using this approach should not trust it blindly. It remains important to always check the actual fairness level of the learned model. On the other hand, we believe that, due to its simplicity and its versatility, FairGrad could be easily deployed in various practical contexts and, thus, could contribute to the dissemination of fair models.

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

## A    RELATED WORK

The fairness literature is extensive and we refer the interested reader to recent surveys (Caton & Haas, 2020; Mehrabi et al., 2021) to get an overview of the subject. Here, we focus on recent works that are more closely related to our approach.

**BiFair (Ozdayi et al., 2021).**    This paper proposes a bilevel optimization scheme for fairness. The idea is to use an outer optimization scheme that learns weights for each example so that the trade-off between fairness and accuracy is as favorable as possible while an inner optimization scheme learns a model that is as accurate as possible. One of the limits of this approach is that it does not directly optimize the fairness level of the model but rather a relaxation that does not provide any guarantees on the goodness of the learned predictor. Furthermore, it is limited to binary classification with binary sensitive attribute. In this paper, we also learn weights for the examples in an iterative way. However, we use a different update rule. Furthermore, we focus on proper fairness definitions rather than relaxations and our objective is to learn accurate models with given levels of fairness rather than a trade-off between the two. Finally, our approach is not limited to the binary setting.

**FairBatch (Roh et al., 2020).**    This paper proposes a batch gradient descent approach that can be used to learn fair models. More precisely, the idea is to draw the batch examples from a skewed distribution that favors the disadvantaged groups by oversampling them. In this paper, we propose to use a reweighting approach which could also be interpreted as altering the distribution of the examples based on their fairness level if all the weights were positive. However, we allow the use of negative weights, and we prove that they are sometimes necessary to achieve fairness. Furthermore, we use a different update rule for the weights.

**AdaFair (Iosifidis & Ntoutsi, 2019).**    This paper proposes a boosting based framework to learn fair models. The underlying idea is to modify the weights of the examples depending on both the performances of the current strong classifier and the group memberships. Hence, examples that belong to the disadvantaged group and are incorrectly classified receive higher weights than the examples that belong to the advantaged group and are correctly classified. In this paper, we use a similar high level idea but we use different weights that do not depend on the performance of the model. Furthermore, rather than a boosting based approach, we consider problems that can be solved using gradient descent. Finally, while AdaFair only focuses on Equalized Odds, we show that our approach works with several fairness notions.

**Identifying and Correcting Label Bias in Machine Learning (Jiang & Nachum, 2020).**    This paper considers the fairness problem from an original point of view as it assumes that the observed labels are biased compared to the true labels. The goal is then to learn a model with respect to the true labels using only the observed labels. To this end, the paper proposes to use an iterative reweighting procedure where positive weights for the examples and updated models are alternatively learned. In this paper, we also propose a reweighting approach. However, we use different weights that are not necessarily positive. Furthermore, our approach is not limited to binary labels and can handle multiclass problems.

## B    REFORMULATION OF VARIOUS GROUP FAIRNESS NOTION

In this section, we present several group fairness notions which respect our fairness definition presented in Section 2.1.

**Example 2** (**Equalized Odds (EOdds) (Hardt et al., 2016)**)**.** A model $h_\theta$ is fair for Equalized Odds when the probability of predicting the correct label is independent of the sensitive attribute,

that is, $\forall l \in \mathcal{Y}, \forall r \in \mathcal{S}$

$$\mathbb{P}\left(h_\theta(x) = l \mid s = r, y = l\right) = \mathbb{P}\left(h_\theta(x) = l \mid y = l\right).$$

It means that we need to partition the space into $K = |\mathcal{Y} \times \mathcal{S}|$ groups and, $\forall l \in \mathcal{Y}, \forall r \in \mathcal{S}$, we define $F_{(l,r)}$ as

$$\begin{aligned}
F_{(l,r)}(\mathcal{T}, h_\theta) &= \mathbb{P}\left(h_\theta(x) \neq l \mid y = l\right) - \mathbb{P}\left(h_\theta(x) \neq l \mid s = r, y = l\right) \\
&= \sum_{(l,r') \neq (l,r)} \mathbb{P}\left(s = r' | y = l\right) \mathbb{P}\left(h_\theta(x) \neq l \mid s = r', y = l\right) \\
&\quad - (1 - \mathbb{P}\left(s = r | y = l\right)) \mathbb{P}\left(h_\theta(x) \neq l \mid s = r, y = l\right)
\end{aligned}$$

where the law of total probability was used to obtain the last equation. Thus, Equalized Odds satisfies all our assumptions with $C_{(l,r)}^{(l,r)} = \mathbb{P}\left(s = r | y = l\right) - 1$, $C_{(l,r)}^{(l,r')} = \mathbb{P}\left(s = r' | y = l\right)$, $C_{(l,r)}^{(l',r')} = 0$ with $r' \neq r$ and $l' \neq l$, and $C_{(l,r)}^0 = 0$.

**Example 3** (**Equality of Opportunity (EOpp) (Hardt et al., 2016)**). A model $h_\theta$ is fair for Equality of Opportunity when the probability of predicting the correct label is independent of the sensitive attribute for a given subset $\mathcal{Y}' \subset \mathcal{Y}$ of labels called the desirable outcomes, that is, $\forall l \in \mathcal{Y}', \forall r \in \mathcal{S}$

$$\mathbb{P}\left(h_\theta(x) = l \mid s = r, y = l\right) = \mathbb{P}\left(h_\theta(x) = l \mid y = l\right).$$

It means that we need to partition the space into $K = |\mathcal{Y} \times \mathcal{S}|$ groups and, $\forall l \in \mathcal{Y}, \forall r \in \mathcal{S}$, we define $F_{(l,r)}$ as

$$F_{(l,r)}(\mathcal{T}, h_\theta) = \left\{ \begin{array}{ll} \begin{array}{l} \mathbb{P}\left(h_\theta(x) = l \mid s = r, y = l\right) \\ \quad - \mathbb{P}\left(h_\theta(x) = l \mid y = l\right) \end{array} & \forall (l,r) \in \mathcal{Y}' \times \mathcal{S} \\ 0 & \forall (l,r) \in \mathcal{Y} \times \mathcal{S} \setminus \mathcal{Y}' \times \mathcal{S} \end{array} \right.$$

which can then be rewritten in the correct form in the same way as Equalized Odds, the only difference being that $C_{(l,r)}^{\cdot} = 0, \forall (l,r) \in \mathcal{Y} \times \mathcal{S} \setminus \mathcal{Y}' \times \mathcal{S}$.

**Example 4** (**Demographic Parity (DP) (Calders et al., 2009)**). A model $h_\theta$ is fair for Demographic Parity when the probability of predicting a binary label is independent of the sensitive attribute, that is, $\forall l \in \mathcal{Y}, \forall r \in \mathcal{S}$

$$\mathbb{P}\left(h_\theta(x) = l \mid s = r\right) = \mathbb{P}\left(h_\theta(x) = l\right).$$

It means that we need to partition the space into $K = |\mathcal{Y} \times \mathcal{S}|$ groups and, $\forall l \in \mathcal{Y}, \forall r \in \mathcal{S}$, we define $F_{(l,r)}$ as

$$\begin{aligned}
F_{(l,r)}(\mathcal{T}, h_\theta) &= \mathbb{P}\left(h_\theta(x) \neq l\right) - \mathbb{P}\left(h_\theta(x) \neq l \mid s = r\right) \\
&= \left(\mathbb{P}\left(y = l, s = r\right) - \mathbb{P}\left(y = l \mid s = r\right)\right) \mathbb{P}\left(h_\theta(x) \neq y \mid s = r, y = l\right) \\
&\quad + \sum_{(l,r') \neq (l,r)} \mathbb{P}\left(y = l, s = r'\right) \mathbb{P}\left(h_\theta(x) \neq y \mid s = r', y = l\right) \\
&\quad + \left(\mathbb{P}\left(y = \bar{l} \mid s = r\right) - \mathbb{P}\left(y = \bar{l}, s = r\right)\right) \mathbb{P}\left(h_\theta(x) \neq y \mid s = r, y = \bar{l}\right) \\
&\quad - \sum_{(\bar{l},r') \neq (\bar{l},r)} \mathbb{P}\left(y = \bar{l}, s = r'\right) \mathbb{P}\left(h_\theta(x) \neq y \mid s = r', y = \bar{l}\right) \\
&\quad \mathbb{P}\left(y = \bar{l}\right) - \mathbb{P}\left(y = \bar{l} \mid s = r\right)
\end{aligned}$$

where the law of total probability was used to obtain the last equation. Thus, Demographic Parity satisfies all our assumptions with $C_{(l,r)}^{(l,r)} = \mathbb{P}\left(y = l, s = r\right) - \mathbb{P}\left(y = l \mid s = r\right)$, $C_{(l,r)}^{(l,r')} = \mathbb{P}\left(y = l, s = r'\right)$ with $r' \neq r$, $C_{(l,r)}^{(\bar{l},r)} = \mathbb{P}\left(y = \bar{l} \mid s = r\right) - \mathbb{P}\left(y = \bar{l}, s = r\right)$, $C_{(l,r)}^{(\bar{l},r')} = -\mathbb{P}\left(y = \bar{l}, s = r'\right)$ with $r' \neq r$, and $C_{(l,r)}^0 = \mathbb{P}\left(y = \bar{l}\right) - \mathbb{P}\left(y = \bar{l} \mid s = r\right)$.

## C  PROOF OF LEMMA 1

**Lemma 2** (Negative weights are necessary.)**.** *Assume that the fairness notion under consideration is Accuracy Parity. Let $h_\theta^*$ be the most accurate and fair model. Then using negative weights is necessary as long as*

$$\min_{\substack{h_\theta \in \mathcal{H} \\ h_\theta \, unfair}} \max_{\mathcal{T}_k} \mathbb{P}\left(h_\theta(x) \neq y | \mathcal{T}_k\right) < \mathbb{P}\left(h_\theta^*(x) \neq y\right).$$

*Proof.* To prove this Lemma, one first need to notice that, for Accuracy Parity, since $\sum_{k=1}^{K} \mathbb{P}(\mathcal{T}_k) = 1$ we have that

$$\sum_{k'=1}^{K} C_k^{k'} = (\mathbb{P}(\mathcal{T}_k) - 1) + \sum_{\substack{k'=1 \\ k' \neq k}}^{K} \mathbb{P}(\mathcal{T}_{k'}) = 0.$$

This implies that

$$\sum_{k=1}^{K} \left[ \mathbb{P}(\mathcal{T}_k) + \sum_{k'=1}^{K} C_{k'}^k \lambda_{k'} \right] = 1.$$

This implies that, whatever our choice of $\lambda$, the weights will always sum to one. In other words, since we also have that $\sum_{k=1}^{K} \lambda_k C_k^0 = 0$ by definition, for a given hypothesis $h_\theta$, we have that

$$\max_{\lambda_1,\ldots,\lambda_K \in \mathbb{R}} \sum_{k=1}^{K} \mathbb{P}(h_\theta(x) \neq y | \mathcal{T}_k) \left[ \mathbb{P}(\mathcal{T}_k) + \sum_{k'=1}^{K} C_{k'}^k \lambda_{k'} \right] \tag{5}$$

$$= \max_{\substack{w_1,\ldots,w_K \in \mathbb{R} \\ s.t. \sum_k w_k = 1}} \sum_{k=1}^{K} \mathbb{P}(h_\theta(x) \neq y | \mathcal{T}_k) w_k \tag{6}$$

where, given $w_1, \ldots, w_K$, the original values of lambda can be obtained by solving the linear system $C\lambda = w$ where

$$C = \begin{pmatrix} C_1^1 & \cdots & C_K^1 \\ \vdots & & \vdots \\ C_1^K & \cdots & C_K^K \end{pmatrix}, \quad \lambda = \begin{pmatrix} \lambda_1 \\ \vdots \\ \lambda_K \end{pmatrix}, \quad w = \begin{pmatrix} w_1 - \mathbb{P}(\mathcal{T}_1) \\ \vdots \\ w_K - \mathbb{P}(\mathcal{T}_K) \end{pmatrix}$$

which is guaranteed to have infinitely many solutions since the rank of the matrix $C$ is $K-1$ and the rank of the augmented matrix $(C|w)$ is also $K-1$. Here we are using the fact that $\mathbb{P}(\mathcal{T}_k) \neq 0, \forall k$ since all the groups have to be represented to be taken into account.

We will now assume that all the weights are positive, that is $w_k \geq 0, \forall k$. Then, the best strategy to solve Problem (6) is to put all the weight on the worst off group $k$, that is set $w_k = 1$ and $w_{k'} = 0, \forall k' \neq k$. It implies that

$$\max_{\substack{w_1,\ldots,w_K \in \mathbb{R} \\ s.t. \sum_k w_k = 1}} \sum_{k=1}^{K} \mathbb{P}(h_\theta(x) \neq y | \mathcal{T}_k) w_k = \max_k \mathbb{P}(h_\theta(x) \neq y | \mathcal{T}_k).$$

Furthermore, notice that, for fair models with respect to Accuracy Parity, we have that $\mathbb{P}(h_\theta(x) \neq y | \mathcal{T}_k) = \mathbb{P}(h_\theta(x) \neq y), \forall k$. Thus, if it holds that

$$\min_{\substack{h_\theta \in \mathcal{H} \\ h_\theta \text{unfair}}} \max_{\mathcal{T}_k} \mathbb{P}(h_\theta(x) \neq y | \mathcal{T}_k) < \mathbb{P}(h_\theta^*(x) \neq y)$$

where $h_\theta^*$ is the most accurate and fair model, then the optimal solution of Problem (3) in the main paper will be unfair. It implies that, in this case, using positive weights is not sufficient and negative weights are necessary. $\qquad\square$

## D  FAIRGRAD FOR $\epsilon$-FAIRNESS

To derive FairGrad for $\epsilon$-fairness we first consider the following standard optimization problem

$$\underset{h_\theta \in \mathcal{H}}{\arg\min} \, \mathbb{P}(h_\theta(x) \neq y)$$

$$\text{s.t. } \forall k \in [K], F_k(\mathcal{T}, h_\theta) \leq \epsilon$$
$$\forall k \in [K], F_k(\mathcal{T}, h_\theta) \geq -\epsilon.$$

We, once again, use a standard multipliers approach to obtain the following unconstrained formulation:

$$\mathcal{L}\left(h_\theta, \lambda_1, \ldots, \lambda_K, \delta_1, \ldots, \delta_K\right) = \mathbb{P}\left(h_\theta(x) \neq y\right) + \sum_{k=1}^{K} \lambda_k\left(F_k(\mathcal{T}, h_\theta) - \epsilon\right) - \delta_k\left(F_k(\mathcal{T}, h_\theta) + \epsilon\right) \tag{7}$$

where $\lambda_1, \ldots, \lambda_K$ and $\delta_1, \ldots, \delta_K$ are the multipliers that belong to $\mathbb{R}^+$, that is the set of positive reals. Once again, to solve this problem, we will use an alternating approach where the hypothesis and the multipliers are updated one after the other.

**Updating the Multipliers.** To update the values $\lambda_1, \ldots, \lambda_K$, we will use a standard gradient ascent procedure. Hence, noting that the gradient of the previous formulation is

$$\nabla_{\lambda_1,\ldots,\lambda_K}\mathcal{L}\left(h_\theta, \lambda_1, \ldots, \lambda_K, \delta_1, \ldots, \delta_K\right) = \begin{pmatrix} F_1(\mathcal{T}, h_\theta) - \epsilon \\ \vdots \\ F_K(\mathcal{T}, h_\theta) - \epsilon \end{pmatrix}$$

$$\nabla_{\delta_1,\ldots,\delta_K}\mathcal{L}\left(h_\theta, \lambda_1, \ldots, \lambda_K, \delta_1, \ldots, \delta_K\right) = \begin{pmatrix} -F_1(\mathcal{T}, h_\theta) - \epsilon \\ \vdots \\ -F_K(\mathcal{T}, h_\theta) - \epsilon \end{pmatrix}$$

we have the following update rule $\forall k \in [K]$

$$\lambda_k^{T+1} = \max\left(0, \lambda_k^T + \eta\left(F_k\left(\mathcal{T}, h_\theta^T\right) - \epsilon\right)\right)$$
$$\delta_k^{T+1} = \max\left(0, \delta_k^T - \eta\left(F_k\left(\mathcal{T}, h_\theta^T\right) + \epsilon\right)\right)$$

where $\eta$ is a learning rate that controls the importance of each weight update.

**Updating the Model.** To update the parameters $\theta \in \mathbb{R}^D$ of the model $h_\theta$, we proceed as before, using a gradient descent approach. However, first, we notice that given the fairness notions that we consider, Equation (7) is equivalent to

$$\mathcal{L}\left(h_\theta, \lambda_1, \ldots, \lambda_K, \delta_1, \ldots, \delta_K\right) = \sum_{k=1}^{K} \mathbb{P}\left(h_\theta(x) \neq y | \mathcal{T}_k\right) \left[\mathbb{P}\left(\mathcal{T}_k\right) + \sum_{k'=1}^{K} C_{k'}^k\left(\lambda_{k'} - \delta_{k'}\right)\right] \tag{8}$$
$$- \sum_{k=1}^{K}\left(\lambda_k + \delta_k\right)\epsilon + \sum_{k=1}^{K}(\lambda_k - \delta_k)C_k^0.$$

Since the additional terms in the optimization problem do not depend on $h_\theta$, the main difference between exact and $\epsilon$-fairness is the nature of the weights. More precisely, at iteration $t$, the update rule becomes

$$\theta^{T+1} = \theta^T - \eta_\theta \sum_{k=1}^{K} \left[\mathbb{P}\left(\mathcal{T}_k\right) + \sum_{k'=1}^{K} C_{k'}^k\left(\lambda_{k'} - \delta_{k'}\right)\right] \nabla_\theta \mathbb{P}\left(h_\theta(x) \neq y | \mathcal{T}_k\right)$$

where $\eta_\theta$ is a learning rate. Once again, we obtain a simple reweighting scheme where the weights depend on the current fairness level of the model through $\lambda_1, \ldots, \lambda_K$ and $\delta_1, \ldots, \delta_K$, the relative size of each group through $\mathbb{P}\left(\mathcal{T}_k\right)$, and the fairness notion through the constants $C$.

# E    EXTENDED EXPERIMENTS

In this section, we provide additional details related to the baselines and the hyper-parameters tuning procedure. We then provide descriptions of the datasets and finally the results.

### E.1 BASELINES

- **Adversarial**: One of the common ways of removing sensitive information from the model's representation is via adversarial learning. Broadly, it consists of three components, namely an encoder, a task classifier, and an adversary. One the on hand, the objective of the adversary is to predict sensitive information from the encoder. On the other hand, the encoder aims to create representations that are useful for the downstream task (task classifier) and, at the same time, fool the adversary. The adversary is generally connected to the encoder via a gradient reversal layer (Ganin & Lempitsky, 2015) which acts like an identity function during the forward pass and scales the loss with a parameter $-\lambda$ during the backward pass. In our setting, the encoder is a Multi-Layer Perceptron with two hidden layers of size $64$ and $128$ respectively, and the task classifier is another Multi-Layer Perceptron with a single hidden layer of size $32$. The adversary is the same as the main task classifier. We use a ReLU as the activation function with the dropout set to $0.2$ and employ batch normalization with default PyTorch parameters. As a part of the hyper-parameter tuning, we did a grid search over $\lambda$, varying it between $0.1$ to $3.0$ with an interval of $0.2$.

- **BiFair (Ozdayi et al., 2021)**: For this baseline, we fix the weight parameter to be of length $8$ as suggested in the code released by the authors. In this fixed setting, we perform a grid search over the following hyper-parameters:

  – Batch Size: 128,256,512
  – Weight Decay: 0.0, 0.001
  – Fairness Loss Weight: 0.5, 1, 2, 4
  – Inner Loop Length: 5, 25, 50

- **Constraints**: We use the implementation available in the TensorFlow Constrained Optimization library with default hyper-parameters.

- **FairBatch**: We use the implementation publicly released by the authors.

- **Weighted ERM**: We reweigh each example in the dataset based on inverse of the proportion of the sensitive group it belongs to.

In our initial experiments, we varied the batch size, and learning rates for both Constraints and FairBatch. However, we found that the default hyper-parameters as specified by the authors result in the best performances. In the spirit of being comparable in terms of hyper-parameter search budget, we also fix all hyper-parameters of FairGrad, apart from the batch size and weight decay. We experiment with two different batch sizes namely, $64$ or $512$ for the standard fairness dataset. Similarly, we also experiment with three weight decay values namely, $0.0$, $0.001$ and $0.01$. Note that we also vary weight decay and batch sizes for FairBatch, Adversarial, Unconstrained, and BiFair approach.

For all our experiments, apart from BiFair, we use Batch Gradient Descent as the optimizer with a learning rate of $0.1$ and a gradient clipping of $0.05$ to avoid exploding gradients. For BiFair, we employ the Adam optimizer as suggested by the authors with a learning rate of $0.001$.

**Hyper-parameters selection procedure.** As mentioned above, all our baselines come with a number of hyper-parameters (learning rates, batch size, weight decay, . . . ) and selecting the best combination is often key to avoid undesirable behaviours such as over-fitting. In this paper, we proceed as follows. First, for each method, we consider all the $X$ possible hyper-parameter combinations and we run the training procedure for 50 epochs for each combination. Then, we retain all the models returned by the last 5 epochs, that is, for a given method, we have $5X$ models and the goal is to select the best one among them. Since we have access to two measures of performance, we can select either the most accurate model, the most fair model, or a trade-off between the two depending on the goal of the practitioner. In this paper, we chose to focus on the third option and we select the model with the lowest fairness score between certain accuracy intervals. More specifically, let $\alpha^*$ be the highest validation accuracy among the $5X$ models, we choose the model with the lowest validation fairness score amongst all models with a validation accuracy in the interval $[\alpha^* - 0.03, \alpha^*]$.

For FairGrad, FairBatch and Unconstrained, we considered 6 hyper-parameters combinations. For BiFair, we considered 72 such combinations, while for Adversarial, there were 90 combinations.

### E.2 DATASETS

Here, we provide additional details on the datasets used in our experiments. We begin by describing the standard fairness datasets for which we follow the pre-processing procedure as described in Lohaus et al. (2020).

- **Adult**[4]: The dataset (Kohavi, 1996) is composed of $45222$ instances, with $14$ features each describing several attributes of a person. The objective is to predict the income of a person (below or above $50k$) while remaining fair with respect to gender (binary in this case). Following the pre-processing step of Wu et al. (2019), only 9 features were used for training.

- **CelebA**[5]: The dataset (Liu et al., 2015) consists of $202,599$ images, along with $40$ binary attributes associated with each image. We use $38$ of these as features while keeping gender as the sensitive attribute and "Smiling" as the class label.

- **Dutch**[6]: The dataset (Žliobaite et al., 2011) is composed of $60,420$ instances with each instance described by $12$ features. We predict "Low Income" or "High Income" as dictated by the occupation as the main classification task and gender as the sensitive attribute.

- **Compas**[7]: The dataset (Larson et al., 2016) contains $6172$ data points, where each data point has $53$ features. The goal is to predict if the defendant will be arrested again within two years of the decision. The sensitive attribute is race, which has been merged into "White" and "Non White" categories.

- **Communities and Crime**[8]: The dataset (Redmond & Baveja, 2002) is composed of $1994$ instances with $128$ features, of which $29$ have been dropped. The objective is to predict the number of violent crimes in the community, with race being the sensitive attribute.

- **German Credit**[9]: The dataset (Dua et al., 2017) consists of $1000$ instances, with each having $20$ attributes. The objective is to predict a person's creditworthiness (binary), with gender being the sensitive attribute.

- **Gaussian**[10]: It is a toy dataset with binary task label and binary sensitive attribute, introduced in Lohaus et al. (2020). It is constructed by drawing points from different Gaussian distributions. We follow the same mechanism as described in Lohaus et al. (2020), and sample $50000$ data points for each class.

- **Adult Folktables**[11]: This dataset (Ding et al., 2021) is an updated version of the original Adult Income dataset. We use California census data with gender as the sensitive attribute. There are $195665$ instances, with $9$ features describing several attributes of a person. We use the same preprocessing step as recommended by the authors.

For all the dataset, we use a $20\%$ of the data as a test set and $80\%$ as a train set. We further divide the train set into two and keep $25\%$ of the training examples as a validation set. For each repetition, we randomly shuffle the data before splitting it, and thus we had unique splits for each random seed. As a last pre-processing step, we centered and scaled each feature independently by substracting the mean and dividing by the standard deviation both of which were estimated on the training set.

**Twitter Sentiment Analysis**[12]: The dataset (Blodgett et al., 2016) consists of $200k$ tweets with binary sensitive attribute (race) and binary sentiment score. We follow the setup proposed by Han et al. (2021) and Elazar & Goldberg (2018) and create bias in the dataset by changing the proportion of each subgroup (race-sentiment) in the training set. With two sentiment classes being happy and sad, and two race classes being AAE and SAE, the training data consists of 40% AAE-happy, 10%

---

[4] https://archive.ics.uci.edu/ml/datasets/adult
[5] https://mmlab.ie.cuhk.edu.hk/projects/CelebA.html
[6] https://sites.google.com/site/conditionaldiscrimination/
[7] https://github.com/propublica/compas-analysis
[8] http://archive.ics.uci.edu/ml/datasets/communities+and+crime
[9] https://archive.ics.uci.edu/ml/datasets/Statlog+%28German+Credit+Data%29
[10] https://github.com/mlohaus/SearchFair/blob/master/examples/get_synthetic_data.py
[11] https://github.com/zykls/folktables
[12] ttp://slanglab.cs.umass.edu/TwitterAAE/

AAE-sad, 10% SAE-happy, and 40% SAE-sad. The test set remains balanced. The tweets are encoded using the DeepMoji (Felbo et al., 2017) encoder with no fine-tuning, which has been pre-trained over millions of tweets to predict their emoji, thereby predicting the sentiment. Note that the train-test splits are pre-defined and thus do not change based on the random seed of the repetition.

## E.3 STANDARD FAIRNESS DATASETS

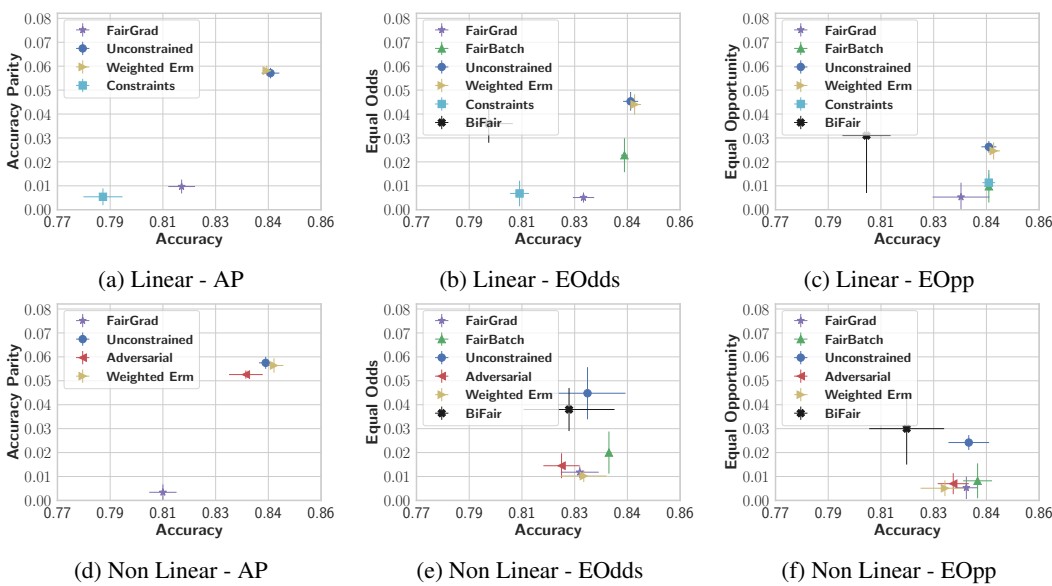

Figure 6: Results for the Adult dataset with different fairness measures.

Table 3: Results for the Adult dataset with Linear Models. All the results are averaged over 5 runs. Here MEAN ABS., MAXIMUM, and MINIMUM represent the mean absolute fairness value, the fairness level of the most well-off group, and the fairness level of the worst-off group, respectively.

| METHOD (L) | ACCURACY ↑ | FAIRNESS | | | |
|---|---|---|---|---|---|
| | | MEASURE | MEAN ABS. ↓ | MAXIMUM | MINIMUM |
| Unconstrained | $0.8456 \pm 0.0033$ | AP | $0.0571 \pm 0.0022$ | $0.077 \pm 0.0029$ | $-0.0373 \pm 0.0017$ |
| Constant | $0.751 \pm 0.0$ | AP | $0.102 \pm 0.0$ | $0.138 \pm 0.0$ | $0.067 \pm 0.0$ |
| Weighted ERM | $0.8442 \pm 0.0016$ | AP | $0.0581 \pm 0.0021$ | $0.0783 \pm 0.0028$ | $-0.0379 \pm 0.0014$ |
| Constrained | $0.783 \pm 0.007$ | AP | $0.005 \pm 0.003$ | $0.007 \pm 0.005$ | $0.004 \pm 0.002$ |
| FairGrad | $0.8124 \pm 0.005$ | AP | $0.0097 \pm 0.0029$ | $0.0131 \pm 0.004$ | $-0.0063 \pm 0.0019$ |
| Unconstrained | $0.846 \pm 0.0028$ | Eodds | $0.0453 \pm 0.0039$ | $0.048 \pm 0.0043$ | $-0.0878 \pm 0.01$ |
| Constant | $0.748 \pm 0.0$ | Eodds | $0.0 \pm 0.0$ | $0.0 \pm 0.0$ | $0.0 \pm 0.0$ |
| Weighted ERM | $0.8475 \pm 0.0024$ | Eodds | $0.044 \pm 0.0043$ | $0.0477 \pm 0.0031$ | $-0.0837 \pm 0.0124$ |
| Constrained | $0.805 \pm 0.004$ | Eodds | $0.007 \pm 0.005$ | $0.019 \pm 0.017$ | $0.002 \pm 0.001$ |
| BiFair | $0.793 \pm 0.009$ | Eodds | $0.036 \pm 0.008$ | $0.085 \pm 0.027$ | $-0.03 \pm 0.016$ |
| FairBatch | $0.8437 \pm 0.0013$ | Eodds | $0.0228 \pm 0.0071$ | $0.0411 \pm 0.0105$ | $-0.0245 \pm 0.0183$ |
| FairGrad | $0.8284 \pm 0.004$ | Eodds | $0.0051 \pm 0.0021$ | $0.0078 \pm 0.0068$ | $-0.0078 \pm 0.0054$ |
| Unconstrained | $0.8457 \pm 0.0028$ | Eopp | $0.0263 \pm 0.0024$ | $0.0157 \pm 0.0011$ | $-0.0893 \pm 0.0083$ |
| Constant | $0.754 \pm 0.0$ | Eopp | $0.0 \pm 0.0$ | $0.0 \pm 0.0$ | $0.0 \pm 0.0$ |
| Weighted ERM | $0.8475 \pm 0.0024$ | Eopp | $0.0246 \pm 0.0036$ | $0.0148 \pm 0.002$ | $-0.0837 \pm 0.0124$ |
| Constrained | $0.846 \pm 0.002$ | Eopp | $0.011 \pm 0.004$ | $0.039 \pm 0.012$ | $0.0 \pm 0.0$ |
| BiFair | $0.8 \pm 0.009$ | Eopp | $0.031 \pm 0.024$ | $0.019 \pm 0.014$ | $-0.107 \pm 0.083$ |
| FairBatch | $0.8457 \pm 0.0016$ | Eopp | $0.0098 \pm 0.0068$ | $0.0225 \pm 0.0174$ | $-0.0166 \pm 0.0241$ |
| FairGrad | $0.8353 \pm 0.0106$ | Eopp | $0.0053 \pm 0.006$ | $0.0177 \pm 0.021$ | $-0.0037 \pm 0.0033$ |

Table 4: Results for the Adult dataset with Non Linear Models. All the results are averaged over 5 runs. Here MEAN ABS., MAXIMUM, and MINIMUM represent the mean absolute fairness value, the fairness level of the most well-off group, and the fairness level of the worst-off group, respectively.

| METHOD (L) | ACCURACY ↑ | FAIRNESS | | | |
|---|---|---|---|---|---|
| | | MEASURE | MEAN ABS. ↓ | MAXIMUM | MINIMUM |
| Unconstrained | $0.8438 \pm 0.0025$ | **AP** | $0.0575 \pm 0.0025$ | $0.0776 \pm 0.0033$ | $-0.0375 \pm 0.0018$ |
| Constant | $0.751 \pm 0.0$ | **AP** | $0.102 \pm 0.0$ | $0.138 \pm 0.0$ | $0.067 \pm 0.0$ |
| Weighted ERM | $0.8469 \pm 0.0035$ | **AP** | $0.0564 \pm 0.003$ | $0.0761 \pm 0.0038$ | $-0.0368 \pm 0.0021$ |
| Adversarial | $0.8364 \pm 0.0063$ | **AP** | $0.0526 \pm 0.0017$ | $0.0709 \pm 0.0025$ | $-0.0343 \pm 0.0009$ |
| FairGrad | $0.8054 \pm 0.0051$ | **AP** | $0.0034 \pm 0.0033$ | $0.0033 \pm 0.0031$ | $-0.0036 \pm 0.0042$ |
| Unconstrained | $0.8299 \pm 0.0142$ | **Eodds** | $0.0448 \pm 0.0109$ | $0.0404 \pm 0.0136$ | $-0.0977 \pm 0.0422$ |
| Constant | $0.748 \pm 0.0$ | **Eodds** | $0.0 \pm 0.0$ | $0.0 \pm 0.0$ | $0.0 \pm 0.0$ |
| Weighted ERM | $0.8285 \pm 0.0085$ | **Eodds** | $0.0102 \pm 0.0025$ | $0.0196 \pm 0.0102$ | $-0.0099 \pm 0.0047$ |
| Adversarial | $0.8202 \pm 0.0068$ | **Eodds** | $0.0145 \pm 0.0052$ | $0.0288 \pm 0.0177$ | $-0.0153 \pm 0.0067$ |
| BiFair | $0.823 \pm 0.017$ | **Eodds** | $0.038 \pm 0.009$ | $0.09 \pm 0.034$ | $-0.038 \pm 0.015$ |
| FairBatch | $0.8379 \pm 0.0009$ | **Eodds** | $0.02 \pm 0.0088$ | $0.0327 \pm 0.0153$ | $-0.0244 \pm 0.0218$ |
| FairGrad | $0.827 \pm 0.0071$ | **Eodds** | $0.0118 \pm 0.0024$ | $0.022 \pm 0.014$ | $-0.0165 \pm 0.0135$ |
| Unconstrained | $0.8382 \pm 0.0076$ | **Eopp** | $0.0242 \pm 0.0031$ | $0.0145 \pm 0.0017$ | $-0.0822 \pm 0.0108$ |
| Constant | $0.754 \pm 0.0$ | **Eopp** | $0.0 \pm 0.0$ | $0.0 \pm 0.0$ | $0.0 \pm 0.0$ |
| Weighted ERM | $0.8293 \pm 0.0091$ | **Eopp** | $0.0051 \pm 0.0033$ | $0.0141 \pm 0.0137$ | $-0.0062 \pm 0.0038$ |
| Adversarial | $0.8324 \pm 0.0058$ | **Eopp** | $0.007 \pm 0.0044$ | $0.0139 \pm 0.0159$ | $-0.0144 \pm 0.0133$ |
| BiFair | $0.815 \pm 0.014$ | **Eopp** | $0.03 \pm 0.015$ | $0.019 \pm 0.009$ | $-0.103 \pm 0.053$ |
| FairBatch | $0.8415 \pm 0.0054$ | **Eopp** | $0.0082 \pm 0.0073$ | $0.0157 \pm 0.0121$ | $-0.017 \pm 0.0271$ |
| FairGrad | $0.8373 \pm 0.0043$ | **Eopp** | $0.0053 \pm 0.0047$ | $0.0099 \pm 0.0146$ | $-0.0112 \pm 0.0127$ |

(a) Linear - AP  (b) Linear - EOdds  (c) Linear - EOpp

(d) Non Linear - AP  (e) Non Linear - EOdds  (f) Non Linear - EOpp

Figure 7: Results for the CelebA dataset with different fairness measures.

.

Table 5: Results for the CelebA dataset with Linear Models. All the results are averaged over 5 runs. Here MEAN ABS., MAXIMUM, and MINIMUM represent the mean absolute fairness value, the fairness level of the most well-off group, and the fairness level of the worst-off group, respectively.

| METHOD (L) | ACCURACY ↑ | FAIRNESS | | | |
|---|---|---|---|---|---|
| | | MEASURE | MEAN ABS. ↓ | MAXIMUM | MINIMUM |
| Unconstrained | 0.8532 ± 0.0009 | **AP** | 0.0204 ± 0.0022 | 0.017 ± 0.0019 | -0.0238 ± 0.0025 |
| Constant | 0.516 ± 0.0 | **AP** | 0.072 ± 0.0 | 0.084 ± 0.0 | 0.06 ± 0.0 |
| Weighted ERM | 0.853 ± 0.0008 | **AP** | 0.0193 ± 0.0021 | 0.0161 ± 0.0018 | -0.0225 ± 0.0023 |
| Constrained | 0.799 ± 0.013 | **AP** | 0.01 ± 0.001 | 0.012 ± 0.002 | 0.009 ± 0.001 |
| FairGrad | 0.835 ± 0.0028 | **AP** | 0.0012 ± 0.0009 | 0.0011 ± 0.0007 | -0.0014 ± 0.0011 |
| Unconstrained | 0.8532 ± 0.0009 | **Eodds** | 0.0499 ± 0.0019 | 0.0538 ± 0.0024 | -0.1011 ± 0.0033 |
| Constant | 0.518 ± 0.0 | **Eodds** | 0.0 ± 0.0 | 0.0 ± 0.0 | 0.0 ± 0.0 |
| Weighted ERM | 0.853 ± 0.0009 | **Eodds** | 0.0504 ± 0.0019 | 0.0532 ± 0.0024 | -0.1001 ± 0.0032 |
| Constrained | 0.802 ± 0.004 | **Eodds** | 0.006 ± 0.001 | 0.01 ± 0.003 | 0.002 ± 0.001 |
| BiFair | 0.845 ± 0.007 | **Eodds** | 0.021 ± 0.005 | 0.02 ± 0.003 | -0.036 ± 0.009 |
| FairBatch | 0.8518 ± 0.0009 | **Eodds** | 0.0226 ± 0.0017 | 0.0218 ± 0.0028 | -0.0411 ± 0.0053 |
| FairGrad | 0.8274 ± 0.002 | **Eodds** | 0.0025 ± 0.0009 | 0.0038 ± 0.0018 | -0.0046 ± 0.0026 |
| Unconstrained | 0.8532 ± 0.0009 | **Eopp** | 0.0387 ± 0.0014 | 0.0538 ± 0.0024 | -0.1011 ± 0.0033 |
| Constant | 0.518 ± 0.0 | **Eopp** | 0.0 ± 0.0 | 0.0 ± 0.0 | 0.0 ± 0.0 |
| Weighted ERM | 0.853 ± 0.0008 | **Eopp** | 0.0383 ± 0.0014 | 0.0531 ± 0.0024 | -0.0999 ± 0.0032 |
| Constrained | 0.834 ± 0.005 | **Eopp** | 0.002 ± 0.001 | 0.005 ± 0.002 | 0.0 ± 0.0 |
| BiFair | 0.848 ± 0.004 | **Eopp** | 0.014 ± 0.006 | 0.02 ± 0.009 | -0.037 ± 0.017 |
| FairBatch | 0.8498 ± 0.001 | **Eopp** | 0.0102 ± 0.0016 | 0.0142 ± 0.0022 | -0.0268 ± 0.0042 |
| FairGrad | 0.844 ± 0.0022 | **Eopp** | 0.0013 ± 0.0009 | 0.0025 ± 0.0021 | -0.0028 ± 0.0018 |

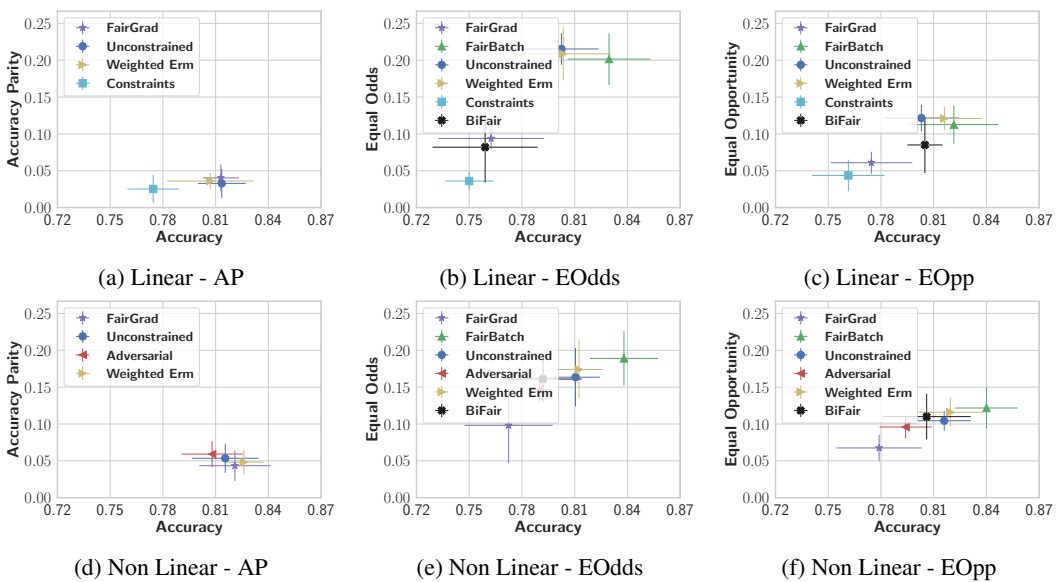

(a) Linear - AP     (b) Linear - EOdds     (c) Linear - EOpp

(d) Non Linear - AP     (e) Non Linear - EOdds     (f) Non Linear - EOpp

Figure 8: Results for the Crime dataset with different fairness measures.

Table 6: Results for the CelebA dataset with Non Linear Models. All the results are averaged over 5 runs. Here MEAN ABS., MAXIMUM, and MINIMUM represent the mean absolute fairness value, the fairness level of the most well-off group, and the fairness level of the worst-off group, respectively.

| METHOD (L) | ACCURACY ↑ | FAIRNESS | | | |
|---|---|---|---|---|---|
| | | MEASURE | MEAN ABS. ↓ | MAXIMUM | MINIMUM |
| Unconstrained | $0.8587 \pm 0.0015$ | AP | $0.0184 \pm 0.0014$ | $0.0154 \pm 0.0012$ | $-0.0215 \pm 0.0016$ |
| Constant | $0.516 \pm 0.0$ | AP | $0.072 \pm 0.0$ | $0.084 \pm 0.0$ | $0.06 \pm 0.0$ |
| Weighted ERM | $0.8593 \pm 0.0018$ | AP | $0.018 \pm 0.0017$ | $0.015 \pm 0.0014$ | $-0.021 \pm 0.0019$ |
| Adversarial | $0.8588 \pm 0.0012$ | AP | $0.0178 \pm 0.0014$ | $0.0148 \pm 0.0012$ | $-0.0208 \pm 0.0015$ |
| FairGrad | $0.8359 \pm 0.0033$ | AP | $0.0023 \pm 0.0012$ | $0.0025 \pm 0.0015$ | $-0.0021 \pm 0.0009$ |
| Unconstrained | $0.8583 \pm 0.0012$ | Eodds | $0.0432 \pm 0.003$ | $0.0475 \pm 0.0028$ | $-0.0893 \pm 0.0049$ |
| Constant | $0.518 \pm 0.0$ | Eodds | $0.0 \pm 0.0$ | $0.0 \pm 0.0$ | $0.0 \pm 0.0$ |
| Weighted ERM | $0.8589 \pm 0.0009$ | Eodds | $0.0419 \pm 0.0021$ | $0.0459 \pm 0.0025$ | $-0.0864 \pm 0.0038$ |
| Adversarial | $0.8567 \pm 0.0014$ | Eodds | $0.0223 \pm 0.002$ | $0.0272 \pm 0.0039$ | $-0.0511 \pm 0.0073$ |
| BiFair | $0.856 \pm 0.004$ | Eodds | $0.023 \pm 0.002$ | $0.028 \pm 0.005$ | $-0.052 \pm 0.009$ |
| FairBatch | $0.8533 \pm 0.0037$ | Eodds | $0.0217 \pm 0.0014$ | $0.0197 \pm 0.0026$ | $-0.0321 \pm 0.005$ |
| FairGrad | $0.8304 \pm 0.0031$ | Eodds | $0.0037 \pm 0.0017$ | $0.0048 \pm 0.0018$ | $-0.0055 \pm 0.0023$ |
| Unconstrained | $0.8585 \pm 0.0016$ | Eopp | $0.0341 \pm 0.002$ | $0.0473 \pm 0.003$ | $-0.0889 \pm 0.0052$ |
| Constant | $0.518 \pm 0.0$ | Eopp | $0.0 \pm 0.0$ | $0.0 \pm 0.0$ | $0.0 \pm 0.0$ |
| Weighted ERM | $0.859 \pm 0.0009$ | Eopp | $0.0331 \pm 0.0014$ | $0.046 \pm 0.0023$ | $-0.0866 \pm 0.0035$ |
| Adversarial | $0.8557 \pm 0.0019$ | Eopp | $0.0161 \pm 0.002$ | $0.0223 \pm 0.0029$ | $-0.0419 \pm 0.0053$ |
| BiFair | $0.854 \pm 0.004$ | Eopp | $0.015 \pm 0.009$ | $0.021 \pm 0.012$ | $-0.039 \pm 0.022$ |
| FairBatch | $0.8475 \pm 0.0043$ | Eopp | $0.0051 \pm 0.0024$ | $0.007 \pm 0.0033$ | $-0.0131 \pm 0.0063$ |
| FairGrad | $0.8439 \pm 0.0063$ | Eopp | $0.0009 \pm 0.0008$ | $0.002 \pm 0.0022$ | $-0.0016 \pm 0.0011$ |

Table 7: Results for the Crime dataset with Linear Models. All the results are averaged over 5 runs. Here MEAN ABS., MAXIMUM, and MINIMUM represent the mean absolute fairness value, the fairness level of the most well-off group, and the fairness level of the worst-off group, respectively.

| METHOD (L) | ACCURACY ↑ | FAIRNESS | | | |
|---|---|---|---|---|---|
| | | MEASURE | MEAN ABS. ↓ | MAXIMUM | MINIMUM |
| Unconstrained | $0.8145 \pm 0.0136$ | AP | $0.0329 \pm 0.0195$ | $0.0258 \pm 0.0162$ | $-0.0399 \pm 0.0229$ |
| Constant | $0.734 \pm 0.0$ | AP | $0.272 \pm 0.0$ | $0.377 \pm 0.0$ | $0.168 \pm 0.0$ |
| Weighted ERM | $0.808 \pm 0.0246$ | AP | $0.0361 \pm 0.0108$ | $0.0284 \pm 0.0091$ | $-0.0438 \pm 0.0129$ |
| Constrained | $0.775 \pm 0.015$ | AP | $0.025 \pm 0.019$ | $0.031 \pm 0.025$ | $0.019 \pm 0.014$ |
| FairGrad | $0.814 \pm 0.0102$ | AP | $0.0403 \pm 0.0181$ | $0.0316 \pm 0.0147$ | $-0.049 \pm 0.0218$ |
| Unconstrained | $0.8035 \pm 0.0212$ | Eodds | $0.2152 \pm 0.0215$ | $0.1038 \pm 0.0231$ | $-0.396 \pm 0.0433$ |
| Constant | $0.677 \pm 0.0$ | Eodds | $0.0 \pm 0.0$ | $0.0 \pm 0.0$ | $0.0 \pm 0.0$ |
| Weighted ERM | $0.8045 \pm 0.0271$ | Eodds | $0.2086 \pm 0.0357$ | $0.0974 \pm 0.0165$ | $-0.3747 \pm 0.0679$ |
| Constrained | $0.751 \pm 0.014$ | Eodds | $0.036 \pm 0.012$ | $0.088 \pm 0.043$ | $0.007 \pm 0.004$ |
| BiFair | $0.76 \pm 0.03$ | Eodds | $0.082 \pm 0.048$ | $0.048 \pm 0.03$ | $-0.163 \pm 0.092$ |
| FairBatch | $0.8306 \pm 0.0237$ | Eodds | $0.2015 \pm 0.035$ | $0.1054 \pm 0.0333$ | $-0.3704 \pm 0.067$ |
| FairGrad | $0.7634 \pm 0.03$ | Eodds | $0.0938 \pm 0.0144$ | $0.0491 \pm 0.016$ | $-0.1927 \pm 0.0362$ |
| Unconstrained | $0.804 \pm 0.0215$ | Eopp | $0.1215 \pm 0.0183$ | $0.1009 \pm 0.0238$ | $-0.3852 \pm 0.0549$ |
| Constant | $0.697 \pm 0.0$ | Eopp | $0.0 \pm 0.0$ | $0.0 \pm 0.0$ | $0.0 \pm 0.0$ |
| Weighted ERM | $0.8171 \pm 0.0213$ | Eopp | $0.1209 \pm 0.0154$ | $0.0985 \pm 0.0106$ | $-0.3851 \pm 0.0599$ |
| Constrained | $0.762 \pm 0.021$ | Eopp | $0.044 \pm 0.021$ | $0.138 \pm 0.066$ | $0.0 \pm 0.0$ |
| BiFair | $0.806 \pm 0.01$ | Eopp | $0.085 \pm 0.038$ | $0.073 \pm 0.042$ | $-0.268 \pm 0.112$ |
| FairBatch | $0.8225 \pm 0.0252$ | Eopp | $0.1126 \pm 0.0259$ | $0.1002 \pm 0.0281$ | $-0.3501 \pm 0.0821$ |
| FairGrad | $0.7755 \pm 0.0233$ | Eopp | $0.0609 \pm 0.0149$ | $0.0507 \pm 0.0166$ | $-0.193 \pm 0.0456$ |

Table 8: Results for the Crime dataset with Non Linear Models. All the results are averaged over 5 runs. Here MEAN ABS., MAXIMUM, and MINIMUM represent the mean absolute fairness value, the fairness level of the most well-off group, and the fairness level of the worst-off group, respectively.

| METHOD (L) | ACCURACY ↑ | FAIRNESS | | | |
|---|---|---|---|---|---|
| | | MEASURE | MEAN ABS. ↓ | MAXIMUM | MINIMUM |
| Unconstrained | $0.8165 \pm 0.019$ | AP | $0.0535 \pm 0.0199$ | $0.0423 \pm 0.0155$ | $-0.0648 \pm 0.0251$ |
| Constant | $0.734 \pm 0.0$ | AP | $0.272 \pm 0.0$ | $0.377 \pm 0.0$ | $0.168 \pm 0.0$ |
| Weighted ERM | $0.8271 \pm 0.0114$ | AP | $0.0483 \pm 0.0167$ | $0.0382 \pm 0.0139$ | $-0.0584 \pm 0.02$ |
| Adversarial | $0.809 \pm 0.0175$ | AP | $0.0592 \pm 0.0173$ | $0.0464 \pm 0.0135$ | $-0.0719 \pm 0.0223$ |
| FairGrad | $0.822 \pm 0.0203$ | AP | $0.0434 \pm 0.0206$ | $0.0341 \pm 0.0162$ | $-0.0526 \pm 0.0252$ |
| Unconstrained | $0.8115 \pm 0.014$ | Eodds | $0.1635 \pm 0.0395$ | $0.0854 \pm 0.014$ | $-0.3326 \pm 0.0649$ |
| Constant | $0.677 \pm 0.0$ | Eodds | $0.0 \pm 0.0$ | $0.0 \pm 0.0$ | $0.0 \pm 0.0$ |
| Weighted ERM | $0.8135 \pm 0.0137$ | Eodds | $0.1739 \pm 0.0394$ | $0.0861 \pm 0.0212$ | $-0.3309 \pm 0.0778$ |
| Adversarial | $0.791 \pm 0.007$ | Eodds | $0.1464 \pm 0.0168$ | $0.0797 \pm 0.0192$ | $-0.3001 \pm 0.0296$ |
| BiFair | $0.793 \pm 0.022$ | Eodds | $0.161 \pm 0.032$ | $0.091 \pm 0.025$ | $-0.339 \pm 0.048$ |
| FairBatch | $0.8391 \pm 0.0195$ | Eodds | $0.189 \pm 0.0368$ | $0.1106 \pm 0.0313$ | $-0.3828 \pm 0.0671$ |
| FairGrad | $0.7734 \pm 0.0251$ | Eodds | $0.0982 \pm 0.0513$ | $0.0511 \pm 0.0179$ | $-0.2016 \pm 0.0771$ |
| Unconstrained | $0.817 \pm 0.0152$ | Eopp | $0.1044 \pm 0.0133$ | $0.0856 \pm 0.0123$ | $-0.3321 \pm 0.0489$ |
| Constant | $0.697 \pm 0.0$ | Eopp | $0.0 \pm 0.0$ | $0.0 \pm 0.0$ | $0.0 \pm 0.0$ |
| Weighted ERM | $0.8205 \pm 0.0184$ | Eopp | $0.1159 \pm 0.0191$ | $0.0955 \pm 0.019$ | $-0.368 \pm 0.0642$ |
| Adversarial | $0.795 \pm 0.0148$ | Eopp | $0.0959 \pm 0.0153$ | $0.0802 \pm 0.0227$ | $-0.3036 \pm 0.042$ |
| BiFair | $0.807 \pm 0.025$ | Eopp | $0.11 \pm 0.031$ | $0.091 \pm 0.031$ | $-0.351 \pm 0.097$ |
| FairBatch | $0.8411 \pm 0.0177$ | Eopp | $0.1217 \pm 0.0277$ | $0.1083 \pm 0.0311$ | $-0.3784 \pm 0.0891$ |
| FairGrad | $0.7799 \pm 0.0243$ | Eopp | $0.0675 \pm 0.0179$ | $0.0556 \pm 0.0147$ | $-0.2143 \pm 0.0592$ |

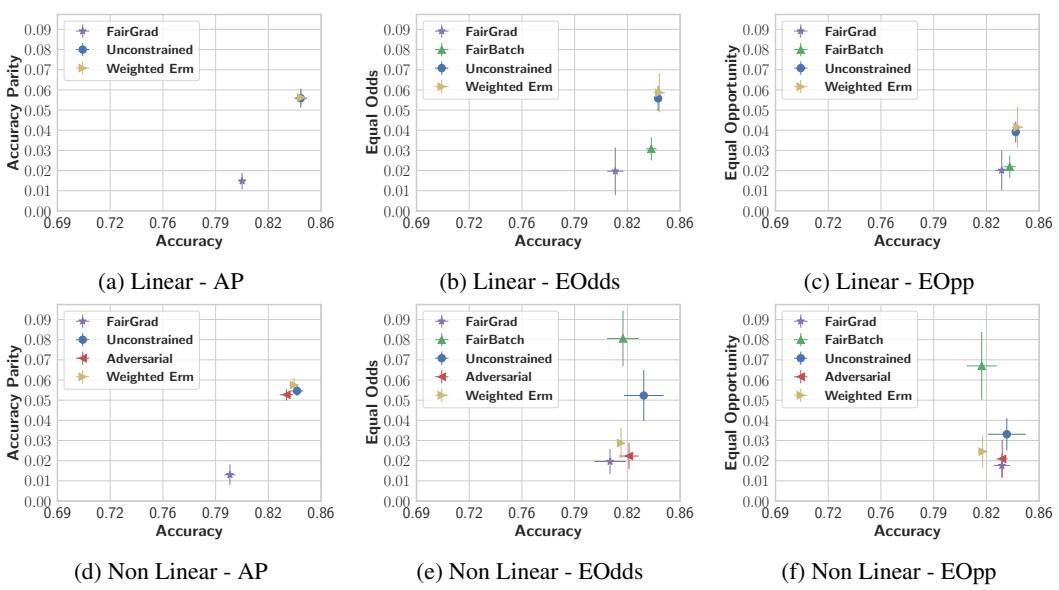

(a) Linear - AP    (b) Linear - EOdds    (c) Linear - EOpp

(d) Non Linear - AP    (e) Non Linear - EOdds    (f) Non Linear - EOpp

Figure 9: Results for the Adult with multiple groups dataset with different fairness measures.

Table 9: Results for the Adult with multiple groups dataset with Linear Models. All the results are averaged over 5 runs. Here MEAN ABS., MAXIMUM, and MINIMUM represent the mean absolute fairness value, the fairness level of the most well-off group, and the fairness level of the worst-off group, respectively.

| METHOD (L) | ACCURACY ↑ | FAIRNESS | | | |
|---|---|---|---|---|---|
| | | MEASURE | MEAN ABS. ↓ | MAXIMUM | MINIMUM |
| Unconstrained | $0.8451 \pm 0.0042$ | AP | $0.0559 \pm 0.0047$ | $0.0985 \pm 0.0111$ | $-0.042 \pm 0.003$ |
| Constant | $0.754 \pm 0.0$ | AP | $0.097 \pm 0.0$ | $0.159 \pm 0.0$ | $0.024 \pm 0.0$ |
| Weighted ERM | $0.8454 \pm 0.0032$ | AP | $0.0562 \pm 0.0042$ | $0.0993 \pm 0.0117$ | $-0.0426 \pm 0.0018$ |
| FairGrad | $0.807 \pm 0.0022$ | AP | $0.0148 \pm 0.0041$ | $0.0256 \pm 0.0048$ | $-0.0107 \pm 0.0045$ |
| Unconstrained | $0.844 \pm 0.0011$ | Eodds | $0.0558 \pm 0.0062$ | $0.0578 \pm 0.0069$ | $-0.1586 \pm 0.0621$ |
| Constant | $0.75 \pm 0.0$ | Eodds | $0.0 \pm 0.0$ | $0.0 \pm 0.0$ | $0.0 \pm 0.0$ |
| Weighted ERM | $0.8448 \pm 0.0038$ | Eodds | $0.0586 \pm 0.0097$ | $0.0567 \pm 0.0048$ | $-0.1702 \pm 0.0776$ |
| FairBatch | $0.8396 \pm 0.0034$ | Eodds | $0.0308 \pm 0.0057$ | $0.0565 \pm 0.0116$ | $-0.0641 \pm 0.0234$ |
| FairGrad | $0.8162 \pm 0.0052$ | Eodds | $0.0197 \pm 0.0118$ | $0.0373 \pm 0.0233$ | $-0.0493 \pm 0.0403$ |
| Unconstrained | $0.8431 \pm 0.002$ | Eopp | $0.0391 \pm 0.0052$ | $0.0297 \pm 0.0131$ | $-0.169 \pm 0.0565$ |
| Constant | $0.762 \pm 0.0$ | Eopp | $0.0 \pm 0.0$ | $0.0 \pm 0.0$ | $0.0 \pm 0.0$ |
| Weighted ERM | $0.8443 \pm 0.0038$ | Eopp | $0.0415 \pm 0.01$ | $0.0316 \pm 0.0145$ | $-0.1767 \pm 0.0797$ |
| FairBatch | $0.8392 \pm 0.004$ | Eopp | $0.0219 \pm 0.0055$ | $0.05 \pm 0.0133$ | $-0.0749 \pm 0.0285$ |
| FairGrad | $0.834 \pm 0.0044$ | Eopp | $0.0201 \pm 0.0099$ | $0.0442 \pm 0.0415$ | $-0.0679 \pm 0.0808$ |

Table 10: Results for the Adult with multiple groups dataset with Non Linear Models. All the results are averaged over 5 runs. Here MEAN ABS., MAXIMUM, and MINIMUM represent the mean absolute fairness value, the fairness level of the most well-off group, and the fairness level of the worst-off group, respectively.

| METHOD (L) | ACCURACY ↑ | FAIRNESS | | | |
|---|---|---|---|---|---|
| | | MEASURE | MEAN ABS. ↓ | MAXIMUM | MINIMUM |
| Unconstrained | $0.8427 \pm 0.0041$ | AP | $0.0546 \pm 0.0026$ | $0.0966 \pm 0.0098$ | $-0.0421 \pm 0.0022$ |
| Constant | $0.754 \pm 0.0$ | AP | $0.097 \pm 0.0$ | $0.159 \pm 0.0$ | $0.024 \pm 0.0$ |
| Weighted ERM | $0.8408 \pm 0.0031$ | AP | $0.0575 \pm 0.0035$ | $0.101 \pm 0.0106$ | $-0.0443 \pm 0.0026$ |
| Adversarial | $0.8358 \pm 0.0043$ | AP | $0.0527 \pm 0.0028$ | $0.0889 \pm 0.0066$ | $-0.0401 \pm 0.0022$ |
| FairGrad | $0.7991 \pm 0.0036$ | AP | $0.013 \pm 0.0051$ | $0.0257 \pm 0.0138$ | $-0.0125 \pm 0.0043$ |
| Unconstrained | $0.8347 \pm 0.0129$ | Eodds | $0.0523 \pm 0.0126$ | $0.0495 \pm 0.0166$ | $-0.1772 \pm 0.0512$ |
| Constant | $0.75 \pm 0.0$ | Eodds | $0.0 \pm 0.0$ | $0.0 \pm 0.0$ | $0.0 \pm 0.0$ |
| Weighted ERM | $0.8199 \pm 0.002$ | Eodds | $0.0287 \pm 0.0076$ | $0.0274 \pm 0.0177$ | $-0.1013 \pm 0.0543$ |
| Adversarial | $0.8251 \pm 0.0064$ | Eodds | $0.0223 \pm 0.0065$ | $0.0451 \pm 0.0308$ | $-0.0667 \pm 0.0559$ |
| FairBatch | $0.8212 \pm 0.0103$ | Eodds | $0.0806 \pm 0.0137$ | $0.0522 \pm 0.0076$ | $-0.2545 \pm 0.0525$ |
| FairGrad | $0.8128 \pm 0.0102$ | Eodds | $0.0196 \pm 0.0061$ | $0.0392 \pm 0.0176$ | $-0.0443 \pm 0.0342$ |
| Unconstrained | $0.8373 \pm 0.0123$ | Eopp | $0.0331 \pm 0.008$ | $0.0183 \pm 0.0045$ | $-0.1587 \pm 0.0643$ |
| Constant | $0.762 \pm 0.0$ | Eopp | $0.0 \pm 0.0$ | $0.0 \pm 0.0$ | $0.0 \pm 0.0$ |
| Weighted ERM | $0.8216 \pm 0.0031$ | Eopp | $0.0245 \pm 0.008$ | $0.0243 \pm 0.0196$ | $-0.1016 \pm 0.0543$ |
| Adversarial | $0.8343 \pm 0.0036$ | Eopp | $0.0209 \pm 0.0093$ | $0.0327 \pm 0.013$ | $-0.0927 \pm 0.0589$ |
| FairBatch | $0.821 \pm 0.0097$ | Eopp | $0.067 \pm 0.0168$ | $0.047 \pm 0.0113$ | $-0.2484 \pm 0.0535$ |
| FairGrad | $0.8341 \pm 0.0053$ | Eopp | $0.0176 \pm 0.0059$ | $0.0302 \pm 0.0272$ | $-0.0731 \pm 0.0543$ |

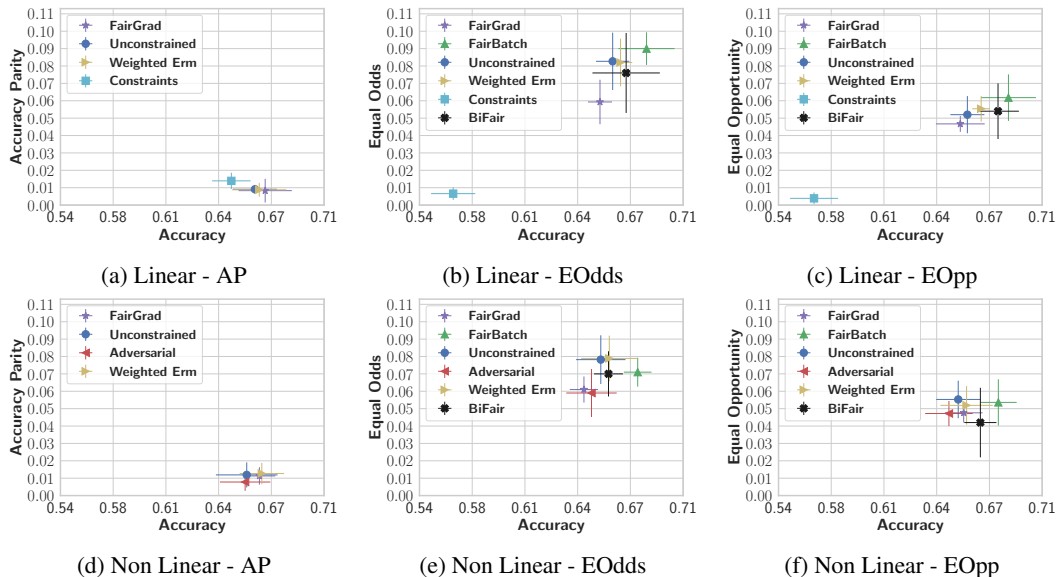

Figure 10: Results for the Compas dataset with different fairness measures.

Table 11: Results for the Compas dataset with Linear Models. All the results are averaged over 5 runs. Here MEAN ABS., MAXIMUM, and MINIMUM represent the mean absolute fairness value, the fairness level of the most well-off group, and the fairness level of the worst-off group, respectively.

| METHOD (L) | ACCURACY ↑ | FAIRNESS | | | |
|---|---|---|---|---|---|
| | | MEASURE | MEAN ABS. ↓ | MAXIMUM | MINIMUM |
| Unconstrained | 0.6644 ± 0.0137 | AP | 0.0091 ± 0.0025 | 0.0076 ± 0.0031 | -0.0107 ± 0.004 |
| Constant | 0.545 ± 0.0 | AP | 0.066 ± 0.0 | 0.085 ± 0.0 | 0.047 ± 0.0 |
| Weighted ERM | 0.6671 ± 0.0169 | AP | 0.0088 ± 0.004 | 0.0061 ± 0.0028 | -0.0115 ± 0.0051 |
| Constrained | 0.65 ± 0.012 | AP | 0.014 ± 0.005 | 0.018 ± 0.006 | 0.009 ± 0.003 |
| FairGrad | 0.6708 ± 0.0166 | AP | 0.0083 ± 0.0068 | 0.0057 ± 0.0048 | -0.0108 ± 0.0088 |
| Unconstrained | 0.6636 ± 0.0104 | Eodds | 0.0827 ± 0.0165 | 0.0758 ± 0.0133 | -0.1553 ± 0.0259 |
| Constant | 0.527 ± 0.0 | Eodds | 0.0 ± 0.0 | 0.0 ± 0.0 | 0.0 ± 0.0 |
| Weighted ERM | 0.6685 ± 0.0073 | Eodds | 0.082 ± 0.0137 | 0.0697 ± 0.0115 | -0.1618 ± 0.0222 |
| Constrained | 0.564 ± 0.014 | Eodds | 0.007 ± 0.004 | 0.014 ± 0.011 | 0.002 ± 0.001 |
| BiFair | 0.672 ± 0.021 | Eodds | 0.076 ± 0.023 | 0.071 ± 0.025 | -0.15 ± 0.039 |
| FairBatch | 0.6847 ± 0.0175 | Eodds | 0.09 ± 0.0094 | 0.0854 ± 0.0149 | -0.1727 ± 0.0304 |
| FairGrad | 0.6557 ± 0.0075 | Eodds | 0.0593 ± 0.0128 | 0.0524 ± 0.0102 | -0.1241 ± 0.0202 |
| Unconstrained | 0.6609 ± 0.0106 | Eopp | 0.052 ± 0.0107 | 0.062 ± 0.0145 | -0.1461 ± 0.0286 |
| Constant | 0.55 ± 0.0 | Eopp | 0.0 ± 0.0 | 0.0 ± 0.0 | 0.0 ± 0.0 |
| Weighted ERM | 0.6695 ± 0.0055 | Eopp | 0.0554 ± 0.0074 | 0.0659 ± 0.0107 | -0.1557 ± 0.0194 |
| Constrained | 0.565 ± 0.015 | Eopp | 0.004 ± 0.003 | 0.011 ± 0.009 | 0.0 ± 0.0 |
| BiFair | 0.68 ± 0.013 | Eopp | 0.054 ± 0.016 | 0.064 ± 0.022 | -0.15 ± 0.044 |
| FairBatch | 0.6865 ± 0.0171 | Eopp | 0.0618 ± 0.0134 | 0.0715 ± 0.0173 | -0.1755 ± 0.0364 |
| FairGrad | 0.6565 ± 0.0152 | Eopp | 0.0467 ± 0.0046 | 0.0554 ± 0.0071 | -0.1313 ± 0.0119 |

Table 12: Results for the Compas dataset with Non Linear Models. All the results are averaged over 5 runs. Here MEAN ABS., MAXIMUM, and MINIMUM represent the mean absolute fairness value, the fairness level of the most well-off group, and the fairness level of the worst-off group, respectively.

| METHOD (L) | ACCURACY ↑ | FAIRNESS | | | |
|---|---|---|---|---|---|
| | | MEASURE | MEAN ABS. ↓ | MAXIMUM | MINIMUM |
| Unconstrained | $0.6593 \pm 0.0192$ | **AP** | $0.0119 \pm 0.0072$ | $0.0095 \pm 0.004$ | $-0.0144 \pm 0.0107$ |
| Constant | $0.545 \pm 0.0$ | **AP** | $0.066 \pm 0.0$ | $0.085 \pm 0.0$ | $0.047 \pm 0.0$ |
| Weighted ERM | $0.6687 \pm 0.0138$ | **AP** | $0.0127 \pm 0.0061$ | $0.011 \pm 0.0034$ | $-0.0145 \pm 0.0099$ |
| Adversarial | $0.6583 \pm 0.0157$ | **AP** | $0.0078 \pm 0.0051$ | $0.0066 \pm 0.0044$ | $-0.009 \pm 0.0069$ |
| FairGrad | $0.6672 \pm 0.0099$ | **AP** | $0.0113 \pm 0.005$ | $0.0095 \pm 0.0023$ | $-0.0131 \pm 0.0082$ |
| Unconstrained | $0.6562 \pm 0.0154$ | **Eodds** | $0.0782 \pm 0.014$ | $0.0715 \pm 0.0136$ | $-0.1521 \pm 0.0277$ |
| Constant | $0.527 \pm 0.0$ | **Eodds** | $0.0 \pm 0.0$ | $0.0 \pm 0.0$ | $0.0 \pm 0.0$ |
| Weighted ERM | $0.6615 \pm 0.0175$ | **Eodds** | $0.0789 \pm 0.0131$ | $0.0726 \pm 0.0077$ | $-0.1496 \pm 0.0313$ |
| Adversarial | $0.6504 \pm 0.0157$ | **Eodds** | $0.059 \pm 0.0138$ | $0.0549 \pm 0.0107$ | $-0.1294 \pm 0.0183$ |
| BiFair | $0.661 \pm 0.009$ | **Eodds** | $0.07 \pm 0.013$ | $0.068 \pm 0.018$ | $-0.133 \pm 0.016$ |
| FairBatch | $0.6792 \pm 0.0086$ | **Eodds** | $0.071 \pm 0.0083$ | $0.0663 \pm 0.0091$ | $-0.1508 \pm 0.0304$ |
| FairGrad | $0.6457 \pm 0.0088$ | **Eodds** | $0.061 \pm 0.0075$ | $0.0564 \pm 0.0065$ | $-0.127 \pm 0.0081$ |
| Unconstrained | $0.6552 \pm 0.0137$ | **Eopp** | $0.0553 \pm 0.0108$ | $0.0659 \pm 0.015$ | $-0.1552 \pm 0.0281$ |
| Constant | $0.55 \pm 0.0$ | **Eopp** | $0.0 \pm 0.0$ | $0.0 \pm 0.0$ | $0.0 \pm 0.0$ |
| Weighted ERM | $0.6604 \pm 0.0163$ | **Eopp** | $0.0519 \pm 0.0111$ | $0.0618 \pm 0.0148$ | $-0.1458 \pm 0.0299$ |
| Adversarial | $0.6494 \pm 0.0148$ | **Eopp** | $0.0472 \pm 0.0072$ | $0.0563 \pm 0.0108$ | $-0.1327 \pm 0.0183$ |
| BiFair | $0.669 \pm 0.01$ | **Eopp** | $0.042 \pm 0.02$ | $0.05 \pm 0.025$ | $-0.117 \pm 0.055$ |
| FairBatch | $0.6802 \pm 0.0114$ | **Eopp** | $0.0536 \pm 0.0133$ | $0.062 \pm 0.0167$ | $-0.1526 \pm 0.0367$ |
| FairGrad | $0.6586 \pm 0.0118$ | **Eopp** | $0.0476 \pm 0.0056$ | $0.0563 \pm 0.0067$ | $-0.1339 \pm 0.0163$ |

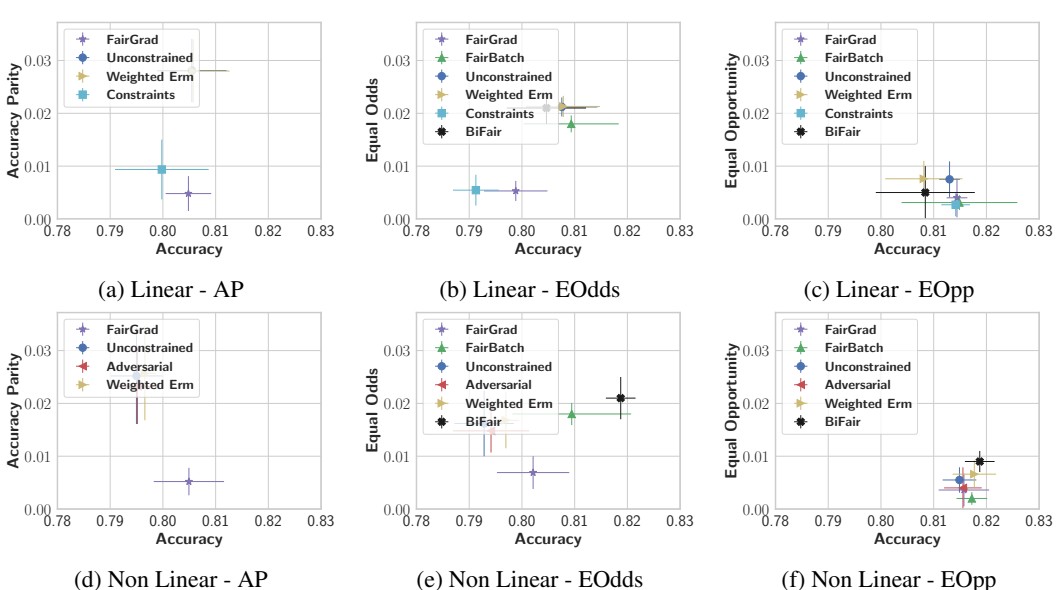

(a) Linear - AP      (b) Linear - EOdds      (c) Linear - EOpp

(d) Non Linear - AP      (e) Non Linear - EOdds      (f) Non Linear - EOpp

Figure 11: Results for the Dutch dataset with different fairness measures.

Table 13: Results for the Dutch dataset with Linear Models. All the results are averaged over 5 runs. Here MEAN ABS., MAXIMUM, and MINIMUM represent the mean absolute fairness value, the fairness level of the most well-off group, and the fairness level of the worst-off group, respectively.

| METHOD (L) | ACCURACY ↑ | FAIRNESS | | | |
|---|---|---|---|---|---|
| | | MEASURE | MEAN ABS. ↓ | MAXIMUM | MINIMUM |
| Unconstrained | $0.8049 \pm 0.007$ | **AP** | $0.0281 \pm 0.006$ | $0.0281 \pm 0.006$ | $-0.0282 \pm 0.0061$ |
| Constant | $0.524 \pm 0.0$ | **AP** | $0.151 \pm 0.0$ | $0.152 \pm 0.0$ | $0.15 \pm 0.0$ |
| Weighted ERM | $0.8052 \pm 0.0073$ | **AP** | $0.028 \pm 0.006$ | $0.028 \pm 0.006$ | $-0.0281 \pm 0.006$ |
| Constrained | $0.799 \pm 0.009$ | **AP** | $0.009 \pm 0.006$ | $0.009 \pm 0.006$ | $0.009 \pm 0.006$ |
| FairGrad | $0.8042 \pm 0.0046$ | **AP** | $0.0048 \pm 0.0033$ | $0.0048 \pm 0.0033$ | $-0.0048 \pm 0.0032$ |
| Unconstrained | $0.8071 \pm 0.0072$ | **Eodds** | $0.0212 \pm 0.0018$ | $0.0322 \pm 0.009$ | $-0.0256 \pm 0.0052$ |
| Constant | $0.522 \pm 0.0$ | **Eodds** | $0.0 \pm 0.0$ | $0.0 \pm 0.0$ | $0.0 \pm 0.0$ |
| Weighted ERM | $0.8074 \pm 0.0074$ | **Eodds** | $0.0213 \pm 0.002$ | $0.032 \pm 0.0086$ | $-0.0254 \pm 0.0051$ |
| Constrained | $0.79 \pm 0.005$ | **Eodds** | $0.005 \pm 0.003$ | $0.009 \pm 0.005$ | $0.002 \pm 0.002$ |
| BiFair | $0.804 \pm 0.008$ | **Eodds** | $0.021 \pm 0.003$ | $0.025 \pm 0.004$ | $-0.033 \pm 0.01$ |
| FairBatch | $0.809 \pm 0.0096$ | **Eodds** | $0.018 \pm 0.0016$ | $0.0262 \pm 0.0039$ | $-0.0211 \pm 0.004$ |
| FairGrad | $0.7978 \pm 0.0064$ | **Eodds** | $0.0053 \pm 0.0019$ | $0.007 \pm 0.0019$ | $-0.009 \pm 0.0049$ |
| Unconstrained | $0.8129 \pm 0.0021$ | **Eopp** | $0.0075 \pm 0.0034$ | $0.0107 \pm 0.0049$ | $-0.0193 \pm 0.0086$ |
| Constant | $0.524 \pm 0.0$ | **Eopp** | $0.0 \pm 0.0$ | $0.0 \pm 0.0$ | $0.0 \pm 0.0$ |
| Weighted ERM | $0.8077 \pm 0.0078$ | **Eopp** | $0.0076 \pm 0.0034$ | $0.011 \pm 0.0049$ | $-0.0196 \pm 0.0087$ |
| Constrained | $0.814 \pm 0.003$ | **Eopp** | $0.003 \pm 0.002$ | $0.007 \pm 0.006$ | $0.0 \pm 0.0$ |
| BiFair | $0.808 \pm 0.01$ | **Eopp** | $0.005 \pm 0.005$ | $0.008 \pm 0.007$ | $-0.012 \pm 0.012$ |
| FairBatch | $0.8149 \pm 0.0117$ | **Eopp** | $0.0031 \pm 0.0014$ | $0.0044 \pm 0.002$ | $-0.0079 \pm 0.0036$ |
| FairGrad | $0.8144 \pm 0.0021$ | **Eopp** | $0.004 \pm 0.0037$ | $0.006 \pm 0.0052$ | $-0.0099 \pm 0.0097$ |

Table 14: Results for the Dutch dataset with Non Linear Models. All the results are averaged over 5 runs. Here MEAN ABS., MAXIMUM, and MINIMUM represent the mean absolute fairness value, the fairness level of the most well-off group, and the fairness level of the worst-off group, respectively.

| METHOD (L) | ACCURACY ↑ | FAIRNESS | | | |
|---|---|---|---|---|---|
| | | MEASURE | MEAN ABS. ↓ | MAXIMUM | MINIMUM |
| Unconstrained | $0.7937 \pm 0.0052$ | **AP** | $0.0252 \pm 0.0091$ | $0.0252 \pm 0.009$ | $-0.0252 \pm 0.0091$ |
| Constant | $0.524 \pm 0.0$ | **AP** | $0.151 \pm 0.0$ | $0.152 \pm 0.0$ | $0.15 \pm 0.0$ |
| Weighted ERM | $0.7954 \pm 0.0023$ | **AP** | $0.0257 \pm 0.0089$ | $0.0257 \pm 0.0089$ | $-0.0257 \pm 0.0089$ |
| Adversarial | $0.7939 \pm 0.0043$ | **AP** | $0.0232 \pm 0.0071$ | $0.0232 \pm 0.0071$ | $-0.0232 \pm 0.007$ |
| FairGrad | $0.8043 \pm 0.0071$ | **AP** | $0.0052 \pm 0.0026$ | $0.0052 \pm 0.0026$ | $-0.0052 \pm 0.0026$ |
| Unconstrained | $0.7914 \pm 0.006$ | **Eodds** | $0.0162 \pm 0.0062$ | $0.0193 \pm 0.0071$ | $-0.0263 \pm 0.0142$ |
| Constant | $0.522 \pm 0.0$ | **Eodds** | $0.0 \pm 0.0$ | $0.0 \pm 0.0$ | $0.0 \pm 0.0$ |
| Weighted ERM | $0.7958 \pm 0.0027$ | **Eodds** | $0.0168 \pm 0.0053$ | $0.0202 \pm 0.0048$ | $-0.0261 \pm 0.0131$ |
| Adversarial | $0.7928 \pm 0.0077$ | **Eodds** | $0.0148 \pm 0.0041$ | $0.0202 \pm 0.0066$ | $-0.0211 \pm 0.006$ |
| BiFair | $0.819 \pm 0.003$ | **Eodds** | $0.021 \pm 0.004$ | $0.03 \pm 0.005$ | $-0.028 \pm 0.007$ |
| FairBatch | $0.8091 \pm 0.012$ | **Eodds** | $0.018 \pm 0.0021$ | $0.0254 \pm 0.0058$ | $-0.0248 \pm 0.0062$ |
| FairGrad | $0.8013 \pm 0.0073$ | **Eodds** | $0.0069 \pm 0.0031$ | $0.0099 \pm 0.0038$ | $-0.0095 \pm 0.0068$ |
| Unconstrained | $0.8149 \pm 0.0034$ | **Eopp** | $0.0055 \pm 0.0024$ | $0.0079 \pm 0.0035$ | $-0.014 \pm 0.0061$ |
| Constant | $0.524 \pm 0.0$ | **Eopp** | $0.0 \pm 0.0$ | $0.0 \pm 0.0$ | $0.0 \pm 0.0$ |
| Weighted ERM | $0.8179 \pm 0.0044$ | **Eopp** | $0.0066 \pm 0.0026$ | $0.0095 \pm 0.0037$ | $-0.017 \pm 0.0065$ |
| Adversarial | $0.8156 \pm 0.0038$ | **Eopp** | $0.004 \pm 0.0039$ | $0.0058 \pm 0.0057$ | $-0.0102 \pm 0.01$ |
| BiFair | $0.819 \pm 0.003$ | **Eopp** | $0.009 \pm 0.002$ | $0.012 \pm 0.003$ | $-0.022 \pm 0.006$ |
| FairBatch | $0.8174 \pm 0.0031$ | **Eopp** | $0.002 \pm 0.0012$ | $0.0029 \pm 0.0017$ | $-0.0052 \pm 0.0031$ |
| FairGrad | $0.8158 \pm 0.0051$ | **Eopp** | $0.0036 \pm 0.0031$ | $0.0051 \pm 0.0045$ | $-0.0092 \pm 0.0079$ |

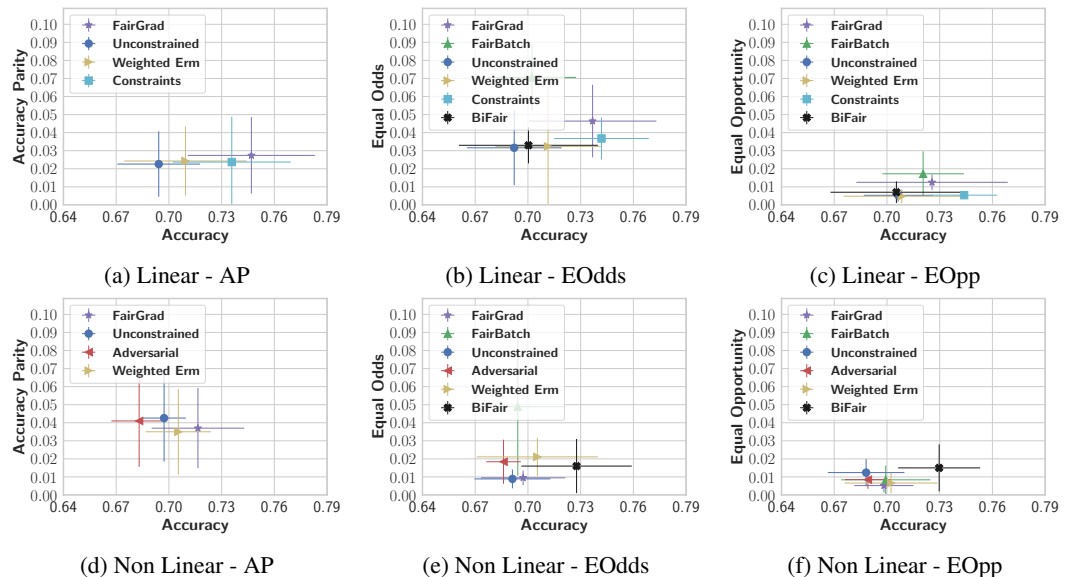

Figure 12: Results for the German dataset with different fairness measures.

Table 15: Results for the German dataset with Linear Models. All the results are averaged over 5 runs. Here MEAN ABS., MAXIMUM, and MINIMUM represent the mean absolute fairness value, the fairness level of the most well-off group, and the fairness level of the worst-off group, respectively.

| METHOD (L) | ACCURACY ↑ | FAIRNESS | | | |
|---|---|---|---|---|---|
| | | MEASURE | MEAN ABS. ↓ | MAXIMUM | MINIMUM |
| Unconstrained | $0.692 \pm 0.0232$ | **AP** | $0.0226 \pm 0.0181$ | $0.0169 \pm 0.0111$ | $-0.0284 \pm 0.0256$ |
| Constant | $0.73 \pm 0.0$ | **AP** | $0.05 \pm 0.0$ | $0.069 \pm 0.0$ | $0.031 \pm 0.0$ |
| Weighted ERM | $0.707 \pm 0.0344$ | **AP** | $0.0243 \pm 0.0191$ | $0.0186 \pm 0.0113$ | $-0.0299 \pm 0.027$ |
| Constrained | $0.733 \pm 0.033$ | **AP** | $0.024 \pm 0.025$ | $0.032 \pm 0.033$ | $0.015 \pm 0.017$ |
| FairGrad | $0.744 \pm 0.0357$ | **AP** | $0.0274 \pm 0.0212$ | $0.0215 \pm 0.0123$ | $-0.0334 \pm 0.0306$ |
| Unconstrained | $0.69 \pm 0.0266$ | **Eodds** | $0.0316 \pm 0.0207$ | $0.0499 \pm 0.0341$ | $-0.0618 \pm 0.0471$ |
| Constant | $0.7 \pm 0.0$ | **Eodds** | $0.0 \pm 0.0$ | $0.0 \pm 0.0$ | $0.0 \pm 0.0$ |
| Weighted ERM | $0.709 \pm 0.0296$ | **Eodds** | $0.0324 \pm 0.0338$ | $0.0461 \pm 0.046$ | $-0.055 \pm 0.0626$ |
| Constrained | $0.739 \pm 0.027$ | **Eodds** | $0.037 \pm 0.012$ | $0.072 \pm 0.025$ | $0.01 \pm 0.004$ |
| BiFair | $0.698 \pm 0.039$ | **Eodds** | $0.033 \pm 0.01$ | $0.052 \pm 0.023$ | $-0.059 \pm 0.029$ |
| FairBatch | $0.7 \pm 0.0247$ | **Eodds** | $0.0706 \pm 0.0184$ | $0.1102 \pm 0.0489$ | $-0.1134 \pm 0.0518$ |
| FairGrad | $0.734 \pm 0.0358$ | **Eodds** | $0.0464 \pm 0.0201$ | $0.0784 \pm 0.0232$ | $-0.0721 \pm 0.0496$ |
| Unconstrained | $0.704 \pm 0.0193$ | **Eopp** | $0.0053 \pm 0.0035$ | $0.0096 \pm 0.004$ | $-0.0116 \pm 0.0117$ |
| Constant | $0.7 \pm 0.0$ | **Eopp** | $0.0 \pm 0.0$ | $0.0 \pm 0.0$ | $0.0 \pm 0.0$ |
| Weighted ERM | $0.706 \pm 0.0328$ | **Eopp** | $0.0048 \pm 0.0039$ | $0.0097 \pm 0.0091$ | $-0.0096 \pm 0.0092$ |
| Constrained | $0.741 \pm 0.019$ | **Eopp** | $0.005 \pm 0.002$ | $0.015 \pm 0.006$ | $0.0 \pm 0.0$ |
| BiFair | $0.703 \pm 0.037$ | **Eopp** | $0.007 \pm 0.006$ | $0.014 \pm 0.015$ | $-0.013 \pm 0.015$ |
| FairBatch | $0.718 \pm 0.0229$ | **Eopp** | $0.0172 \pm 0.0124$ | $0.0272 \pm 0.0187$ | $-0.0416 \pm 0.0396$ |
| FairGrad | $0.723 \pm 0.0425$ | **Eopp** | $0.0125 \pm 0.0043$ | $0.0212 \pm 0.0087$ | $-0.0288 \pm 0.0162$ |

Table 16: Results for the German dataset with Non Linear Models. All the results are averaged over 5 runs. Here MEAN ABS., MAXIMUM, and MINIMUM represent the mean absolute fairness value, the fairness level of the most well-off group, and the fairness level of the worst-off group, respectively.

| METHOD (L) | ACCURACY ↑ | FAIRNESS | | | |
|---|---|---|---|---|---|
| | | MEASURE | MEAN ABS. ↓ | MAXIMUM | MINIMUM |
| Unconstrained | 0.695 ± 0.0122 | AP | 0.0426 ± 0.0241 | 0.0314 ± 0.0144 | -0.0537 ± 0.0345 |
| Constant | 0.73 ± 0.0 | AP | 0.05 ± 0.0 | 0.069 ± 0.0 | 0.031 ± 0.0 |
| Weighted ERM | 0.703 ± 0.0183 | AP | 0.035 ± 0.0237 | 0.0265 ± 0.0138 | -0.0436 ± 0.0338 |
| Adversarial | 0.681 ± 0.0156 | AP | 0.041 ± 0.0254 | 0.0327 ± 0.0165 | -0.0492 ± 0.0368 |
| FairGrad | 0.714 ± 0.026 | AP | 0.037 ± 0.0222 | 0.0291 ± 0.0119 | -0.0448 ± 0.0331 |
| Unconstrained | 0.689 ± 0.0213 | Eodds | 0.0089 ± 0.0052 | 0.0117 ± 0.0045 | -0.0144 ± 0.0116 |
| Constant | 0.7 ± 0.0 | Eodds | 0.0 ± 0.0 | 0.0 ± 0.0 | 0.0 ± 0.0 |
| Weighted ERM | 0.703 ± 0.034 | Eodds | 0.0211 ± 0.0106 | 0.0305 ± 0.0186 | -0.0372 ± 0.0158 |
| Adversarial | 0.684 ± 0.0097 | Eodds | 0.0184 ± 0.0122 | 0.0263 ± 0.0201 | -0.0339 ± 0.0237 |
| BiFair | 0.725 ± 0.031 | Eodds | 0.016 ± 0.015 | 0.021 ± 0.018 | -0.027 ± 0.018 |
| FairBatch | 0.692 ± 0.026 | Eodds | 0.0489 ± 0.0382 | 0.0607 ± 0.0446 | -0.0882 ± 0.0983 |
| FairGrad | 0.695 ± 0.0237 | Eodds | 0.0095 ± 0.004 | 0.0121 ± 0.0046 | -0.0175 ± 0.0076 |
| Unconstrained | 0.686 ± 0.0215 | Eopp | 0.0124 ± 0.0075 | 0.0227 ± 0.0128 | -0.0269 ± 0.0227 |
| Constant | 0.7 ± 0.0 | Eopp | 0.0 ± 0.0 | 0.0 ± 0.0 | 0.0 ± 0.0 |
| Weighted ERM | 0.7 ± 0.0261 | Eopp | 0.0066 ± 0.0057 | 0.0131 ± 0.0071 | -0.0133 ± 0.0173 |
| Adversarial | 0.687 ± 0.0129 | Eopp | 0.0085 ± 0.0051 | 0.0203 ± 0.0147 | -0.0137 ± 0.0099 |
| BiFair | 0.727 ± 0.023 | Eopp | 0.015 ± 0.013 | 0.023 ± 0.019 | -0.036 ± 0.038 |
| FairBatch | 0.697 ± 0.025 | Eopp | 0.0084 ± 0.0079 | 0.0235 ± 0.0226 | -0.0102 ± 0.0094 |
| FairGrad | 0.696 ± 0.0166 | Eopp | 0.0052 ± 0.0038 | 0.0093 ± 0.0064 | -0.0115 ± 0.0108 |

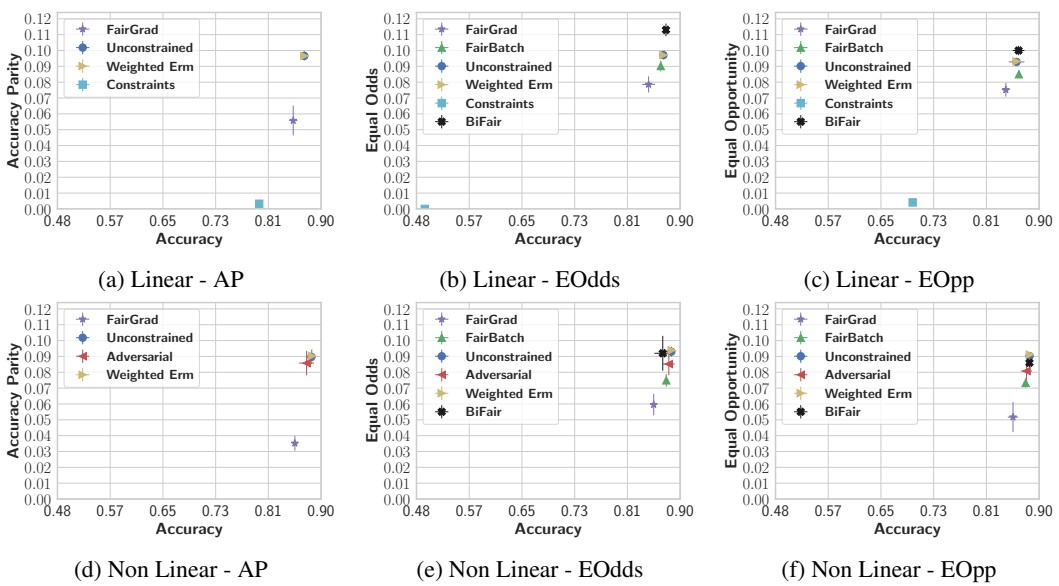

(a) Linear - AP     (b) Linear - EOdds     (c) Linear - EOpp

(d) Non Linear - AP     (e) Non Linear - EOdds     (f) Non Linear - EOpp

Figure 13: Results for the Gaussian dataset with different fairness measures.

Table 17: Results for the Gaussian dataset with Linear Models. All the results are averaged over 5 runs. Here MEAN ABS., MAXIMUM, and MINIMUM represent the mean absolute fairness value, the fairness level of the most well-off group, and the fairness level of the worst-off group, respectively.

| METHOD (L) | ACCURACY ↑ | FAIRNESS | | | |
|---|---|---|---|---|---|
| | | MEASURE | MEAN ABS. ↓ | MAXIMUM | MINIMUM |
| Unconstrained | $0.8689 \pm 0.0037$ | **AP** | $0.0966 \pm 0.0029$ | $0.0957 \pm 0.0028$ | $-0.0974 \pm 0.0036$ |
| Constant | $0.497 \pm 0.0$ | **AP** | $0.001 \pm 0.0$ | $0.001 \pm 0.0$ | $0.001 \pm 0.0$ |
| Weighted ERM | $0.869 \pm 0.0039$ | **AP** | $0.0966 \pm 0.0026$ | $0.0957 \pm 0.0023$ | $-0.0974 \pm 0.0034$ |
| Constrained | $0.799 \pm 0.004$ | **AP** | $0.003 \pm 0.002$ | $0.003 \pm 0.002$ | $0.003 \pm 0.002$ |
| FairGrad | $0.8516 \pm 0.0064$ | **AP** | $0.0558 \pm 0.0094$ | $0.0553 \pm 0.0093$ | $-0.0562 \pm 0.0096$ |
| Unconstrained | $0.869 \pm 0.0037$ | **Eodds** | $0.0971 \pm 0.0026$ | $0.1872 \pm 0.0067$ | $-0.1896 \pm 0.0056$ |
| Constant | $0.499 \pm 0.0$ | **Eodds** | $0.0 \pm 0.0$ | $0.0 \pm 0.0$ | $0.0 \pm 0.0$ |
| Weighted ERM | $0.869 \pm 0.0039$ | **Eodds** | $0.0971 \pm 0.0023$ | $0.1869 \pm 0.0063$ | $-0.1894 \pm 0.0051$ |
| Constrained | $0.497 \pm 0.003$ | **Eodds** | $0.0 \pm 0.0$ | $0.0 \pm 0.0$ | $0.0 \pm 0.0$ |
| BiFair | $0.873 \pm 0.004$ | **Eodds** | $0.113 \pm 0.004$ | $0.21 \pm 0.007$ | $-0.213 \pm 0.004$ |
| FairBatch | $0.8649 \pm 0.0025$ | **Eodds** | $0.0902 \pm 0.0035$ | $0.1717 \pm 0.0046$ | $-0.1719 \pm 0.0079$ |
| FairGrad | $0.8459 \pm 0.01$ | **Eodds** | $0.0786 \pm 0.0051$ | $0.1504 \pm 0.0102$ | $-0.1527 \pm 0.0142$ |
| Unconstrained | $0.8598 \pm 0.0121$ | **Eopp** | $0.0928 \pm 0.0012$ | $0.1845 \pm 0.0041$ | $-0.1869 \pm 0.0041$ |
| Constant | $0.498 \pm 0.0$ | **Eopp** | $0.0 \pm 0.0$ | $0.0 \pm 0.0$ | $0.0 \pm 0.0$ |
| Weighted ERM | $0.8599 \pm 0.0121$ | **Eopp** | $0.0931 \pm 0.0011$ | $0.1849 \pm 0.004$ | $-0.1874 \pm 0.004$ |
| Constrained | $0.698 \pm 0.005$ | **Eopp** | $0.004 \pm 0.002$ | $0.008 \pm 0.005$ | $0.0 \pm 0.0$ |
| BiFair | $0.863 \pm 0.009$ | **Eopp** | $0.1 \pm 0.003$ | $0.2 \pm 0.007$ | $-0.202 \pm 0.006$ |
| FairBatch | $0.8635 \pm 0.0024$ | **Eopp** | $0.085 \pm 0.0023$ | $0.17 \pm 0.0032$ | $-0.1702 \pm 0.0065$ |
| FairGrad | $0.8431 \pm 0.0065$ | **Eopp** | $0.0752 \pm 0.0043$ | $0.1494 \pm 0.0087$ | $-0.1514 \pm 0.0094$ |

Table 18: Results for the Gaussian dataset with Non Linear Models. All the results are averaged over 5 runs. Here MEAN ABS., MAXIMUM, and MINIMUM represent the mean absolute fairness value, the fairness level of the most well-off group, and the fairness level of the worst-off group, respectively.

| METHOD (L) | ACCURACY ↑ | FAIRNESS | | | |
|---|---|---|---|---|---|
| | | MEASURE | MEAN ABS. ↓ | MAXIMUM | MINIMUM |
| Unconstrained | $0.88 \pm 0.0038$ | **AP** | $0.0897 \pm 0.0045$ | $0.0888 \pm 0.0035$ | $-0.0905 \pm 0.0055$ |
| Constant | $0.497 \pm 0.0$ | **AP** | $0.001 \pm 0.0$ | $0.001 \pm 0.0$ | $0.001 \pm 0.0$ |
| Weighted ERM | $0.8809 \pm 0.0048$ | **AP** | $0.0903 \pm 0.0045$ | $0.0894 \pm 0.0033$ | $-0.0911 \pm 0.0057$ |
| Adversarial | $0.8725 \pm 0.0115$ | **AP** | $0.0858 \pm 0.0077$ | $0.0851 \pm 0.0076$ | $-0.0866 \pm 0.0081$ |
| FairGrad | $0.8542 \pm 0.0047$ | **AP** | $0.0352 \pm 0.0047$ | $0.0349 \pm 0.0048$ | $-0.0355 \pm 0.0046$ |
| Unconstrained | $0.8814 \pm 0.0024$ | **Eodds** | $0.093 \pm 0.0032$ | $0.1807 \pm 0.0066$ | $-0.183 \pm 0.005$ |
| Constant | $0.499 \pm 0.0$ | **Eodds** | $0.0 \pm 0.0$ | $0.0 \pm 0.0$ | $0.0 \pm 0.0$ |
| Weighted ERM | $0.8821 \pm 0.0031$ | **Eodds** | $0.0939 \pm 0.0013$ | $0.1826 \pm 0.0042$ | $-0.185 \pm 0.0033$ |
| Adversarial | $0.8775 \pm 0.0091$ | **Eodds** | $0.0852 \pm 0.007$ | $0.1643 \pm 0.0125$ | $-0.1666 \pm 0.0146$ |
| BiFair | $0.868 \pm 0.013$ | **Eodds** | $0.092 \pm 0.011$ | $0.167 \pm 0.035$ | $-0.168 \pm 0.031$ |
| FairBatch | $0.8735 \pm 0.0032$ | **Eodds** | $0.0749 \pm 0.0041$ | $0.1455 \pm 0.0059$ | $-0.1456 \pm 0.0056$ |
| FairGrad | $0.8539 \pm 0.0056$ | **Eodds** | $0.0596 \pm 0.0068$ | $0.1013 \pm 0.0147$ | $-0.1025 \pm 0.0144$ |
| Unconstrained | $0.8801 \pm 0.004$ | **Eopp** | $0.0902 \pm 0.0017$ | $0.1792 \pm 0.0041$ | $-0.1816 \pm 0.0053$ |
| Constant | $0.498 \pm 0.0$ | **Eopp** | $0.0 \pm 0.0$ | $0.0 \pm 0.0$ | $0.0 \pm 0.0$ |
| Weighted ERM | $0.8805 \pm 0.0046$ | **Eopp** | $0.0912 \pm 0.0008$ | $0.1812 \pm 0.0024$ | $-0.1837 \pm 0.0045$ |
| Adversarial | $0.8754 \pm 0.0086$ | **Eopp** | $0.0808 \pm 0.0066$ | $0.1605 \pm 0.0128$ | $-0.1628 \pm 0.0143$ |
| BiFair | $0.88 \pm 0.003$ | **Eopp** | $0.086 \pm 0.005$ | $0.17 \pm 0.013$ | $-0.172 \pm 0.009$ |
| FairBatch | $0.874 \pm 0.0035$ | **Eopp** | $0.0733 \pm 0.0029$ | $0.1465 \pm 0.0054$ | $-0.1467 \pm 0.0066$ |
| FairGrad | $0.8543 \pm 0.0082$ | **Eopp** | $0.0517 \pm 0.0095$ | $0.1028 \pm 0.0191$ | $-0.1041 \pm 0.0192$ |

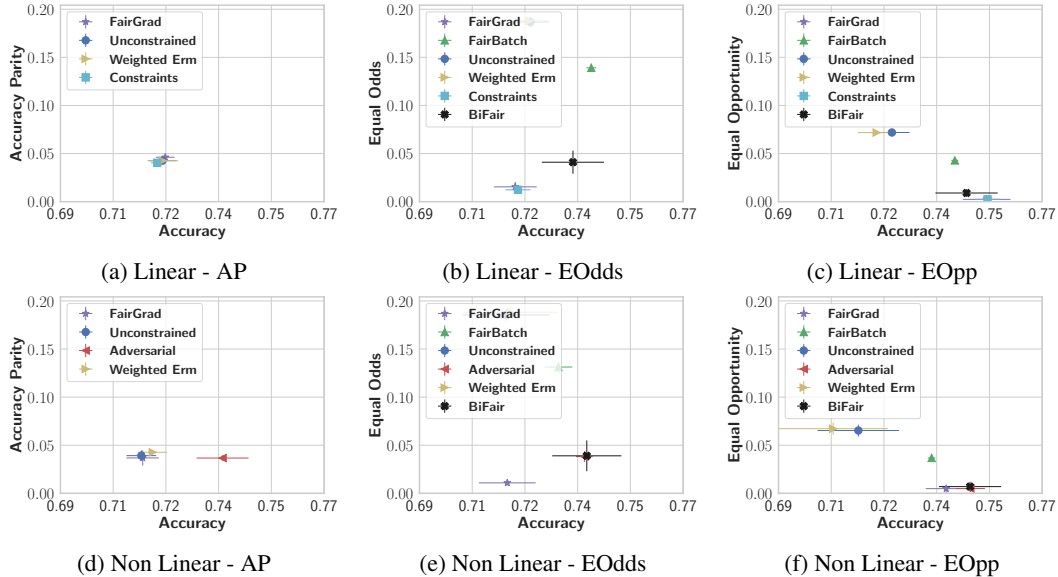

Figure 14: Results for the Twitter Sentiment dataset with different fairness measures.

Table 19: Results for the Twitter Sentiment dataset with Linear Models. All the results are averaged over 5 runs. Here MEAN ABS., MAXIMUM, and MINIMUM represent the mean absolute fairness value, the fairness level of the most well-off group, and the fairness level of the worst-off group, respectively.

| METHOD (L) | ACCURACY ↑ | FAIRNESS | | | |
|---|---|---|---|---|---|
| | | MEASURE | MEAN ABS. ↓ | MAXIMUM | MINIMUM |
| Unconstrained | $0.7211 \pm 0.004$ | **AP** | $0.0426 \pm 0.0011$ | $0.0426 \pm 0.0011$ | $-0.0426 \pm 0.0011$ |
| Constant | $0.5 \pm 0.0$ | **AP** | $0.0 \pm 0.0$ | $0.0 \pm 0.0$ | $0.0 \pm 0.0$ |
| Weighted ERM | $0.7212 \pm 0.0044$ | **AP** | $0.0426 \pm 0.0011$ | $0.0426 \pm 0.0011$ | $-0.0426 \pm 0.0011$ |
| Constrained | $0.72 \pm 0.002$ | **AP** | $0.04 \pm 0.003$ | $0.04 \pm 0.003$ | $0.04 \pm 0.003$ |
| FairGrad | $0.7219 \pm 0.0027$ | **AP** | $0.0462 \pm 0.0021$ | $0.0462 \pm 0.0021$ | $-0.0462 \pm 0.0021$ |
| Unconstrained | $0.7237 \pm 0.0054$ | **Eodds** | $0.1867 \pm 0.0052$ | $0.2287 \pm 0.0078$ | $-0.2288 \pm 0.0078$ |
| Constant | $0.5 \pm 0.0$ | **Eodds** | $0.0 \pm 0.0$ | $0.0 \pm 0.0$ | $0.0 \pm 0.0$ |
| Weighted ERM | $0.7234 \pm 0.0054$ | **Eodds** | $0.188 \pm 0.0033$ | $0.2314 \pm 0.0056$ | $-0.2315 \pm 0.0056$ |
| Constrained | $0.72 \pm 0.004$ | **Eodds** | $0.012 \pm 0.002$ | $0.019 \pm 0.005$ | $0.006 \pm 0.005$ |
| BiFair | $0.736 \pm 0.009$ | **Eodds** | $0.041 \pm 0.012$ | $0.056 \pm 0.022$ | $-0.056 \pm 0.022$ |
| FairBatch | $0.7413 \pm 0.0014$ | **Eodds** | $0.1391 \pm 0.0043$ | $0.1755 \pm 0.0084$ | $-0.1756 \pm 0.0084$ |
| FairGrad | $0.7193 \pm 0.0062$ | **Eodds** | $0.0154 \pm 0.0051$ | $0.0204 \pm 0.0098$ | $-0.0204 \pm 0.0098$ |
| Unconstrained | $0.7244 \pm 0.0051$ | **Eopp** | $0.0719 \pm 0.0012$ | $0.1439 \pm 0.0023$ | $-0.1438 \pm 0.0023$ |
| Constant | $0.5 \pm 0.0$ | **Eopp** | $0.0 \pm 0.0$ | $0.0 \pm 0.0$ | $0.0 \pm 0.0$ |
| Weighted ERM | $0.72 \pm 0.0054$ | **Eopp** | $0.0718 \pm 0.0013$ | $0.1437 \pm 0.0026$ | $-0.1436 \pm 0.0026$ |
| Constrained | $0.752 \pm 0.004$ | **Eopp** | $0.002 \pm 0.001$ | $0.005 \pm 0.001$ | $0.0 \pm 0.0$ |
| BiFair | $0.746 \pm 0.009$ | **Eopp** | $0.009 \pm 0.004$ | $0.017 \pm 0.009$ | $-0.017 \pm 0.009$ |
| FairBatch | $0.7426 \pm 0.001$ | **Eopp** | $0.0429 \pm 0.0005$ | $0.0858 \pm 0.0011$ | $-0.0858 \pm 0.0011$ |
| FairGrad | $0.7518 \pm 0.0069$ | **Eopp** | $0.0024 \pm 0.002$ | $0.0049 \pm 0.004$ | $-0.0049 \pm 0.004$ |

Table 20: Results for the Twitter Sentiment dataset with Non Linear Models. All the results are averaged over 5 runs. Here MEAN ABS., MAXIMUM, and MINIMUM represent the mean absolute fairness value, the fairness level of the most well-off group, and the fairness level of the worst-off group, respectively.

| METHOD (L) | ACCURACY ↑ | | FAIRNESS | | |
| --- | --- | --- | --- | --- | --- |
| | | MEASURE | MEAN ABS. ↓ | MAXIMUM | MINIMUM |
| Unconstrained | $0.715 \pm 0.0043$ | **AP** | $0.0392 \pm 0.0055$ | $0.0392 \pm 0.0055$ | $-0.0392 \pm 0.0055$ |
| Constant | $0.5 \pm 0.0$ | **AP** | $0.0 \pm 0.0$ | $0.0 \pm 0.0$ | $0.0 \pm 0.0$ |
| Weighted ERM | $0.7183 \pm 0.0042$ | **AP** | $0.0427 \pm 0.0019$ | $0.0427 \pm 0.0019$ | $-0.0427 \pm 0.0019$ |
| Adversarial | $0.7385 \pm 0.0075$ | **AP** | $0.0367 \pm 0.0027$ | $0.0367 \pm 0.0027$ | $-0.0368 \pm 0.0027$ |
| FairGrad | $0.7154 \pm 0.0047$ | **AP** | $0.0368 \pm 0.0079$ | $0.0367 \pm 0.0078$ | $-0.0368 \pm 0.0079$ |
| Unconstrained | $0.7167 \pm 0.0126$ | **Eodds** | $0.1854 \pm 0.0061$ | $0.2349 \pm 0.0091$ | $-0.235 \pm 0.0091$ |
| Constant | $0.5 \pm 0.0$ | **Eodds** | $0.0 \pm 0.0$ | $0.0 \pm 0.0$ | $0.0 \pm 0.0$ |
| Weighted ERM | $0.718 \pm 0.0137$ | **Eodds** | $0.1882 \pm 0.0062$ | $0.2379 \pm 0.0073$ | $-0.2381 \pm 0.0073$ |
| Adversarial | $0.7393 \pm 0.0024$ | **Eodds** | $0.0382 \pm 0.0056$ | $0.06 \pm 0.0151$ | $-0.06 \pm 0.0151$ |
| BiFair | $0.74 \pm 0.01$ | **Eodds** | $0.039 \pm 0.016$ | $0.058 \pm 0.017$ | $-0.058 \pm 0.017$ |
| FairBatch | $0.7318 \pm 0.004$ | **Eodds** | $0.1313 \pm 0.0057$ | $0.1724 \pm 0.0055$ | $-0.1725 \pm 0.0055$ |
| FairGrad | $0.717 \pm 0.0082$ | **Eodds** | $0.0109 \pm 0.0027$ | $0.0165 \pm 0.0053$ | $-0.0165 \pm 0.0053$ |
| Unconstrained | $0.7147 \pm 0.0118$ | **Eopp** | $0.0653 \pm 0.0062$ | $0.1306 \pm 0.0124$ | $-0.1306 \pm 0.0124$ |
| Constant | $0.5 \pm 0.0$ | **Eopp** | $0.0 \pm 0.0$ | $0.0 \pm 0.0$ | $0.0 \pm 0.0$ |
| Weighted ERM | $0.7074 \pm 0.0158$ | **Eopp** | $0.0672 \pm 0.0062$ | $0.1346 \pm 0.0125$ | $-0.1345 \pm 0.0125$ |
| Adversarial | $0.7471 \pm 0.0042$ | **Eopp** | $0.005 \pm 0.0035$ | $0.0099 \pm 0.007$ | $-0.0099 \pm 0.007$ |
| BiFair | $0.747 \pm 0.009$ | **Eopp** | $0.007 \pm 0.005$ | $0.013 \pm 0.01$ | $-0.013 \pm 0.01$ |
| FairBatch | $0.7359 \pm 0.0011$ | **Eopp** | $0.0368 \pm 0.0012$ | $0.0736 \pm 0.0025$ | $-0.0736 \pm 0.0025$ |
| FairGrad | $0.7401 \pm 0.0059$ | **Eopp** | $0.0049 \pm 0.0041$ | $0.0099 \pm 0.0083$ | $-0.0099 \pm 0.0083$ |

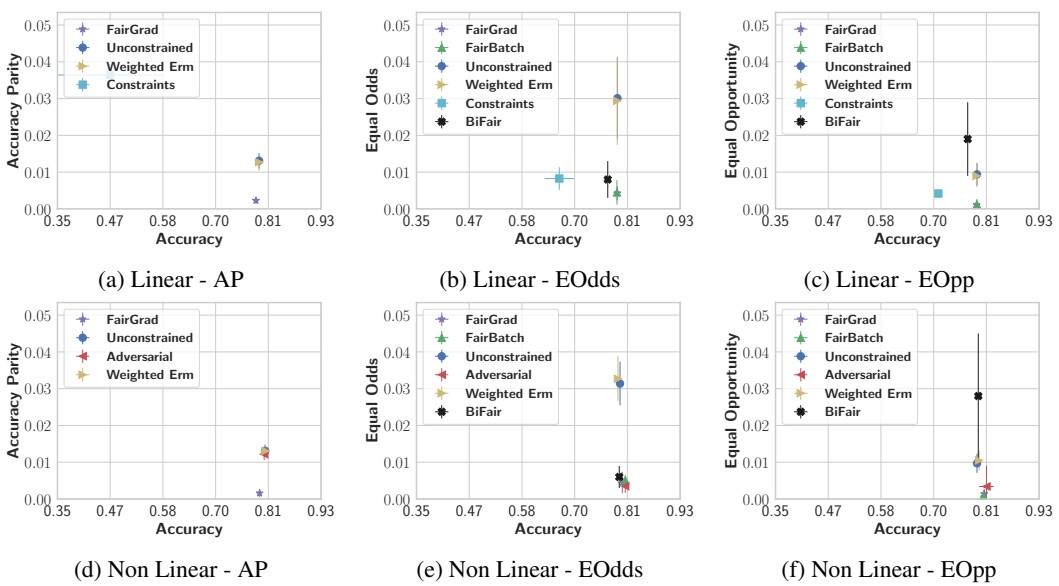

(a) Linear - AP  (b) Linear - EOdds  (c) Linear - EOpp

(d) Non Linear - AP  (e) Non Linear - EOdds  (f) Non Linear - EOpp

Figure 15: Results for the Folktables Adult dataset with different fairness measures.

Table 21: Results for the Folktables Adult dataset with Linear Models. All the results are averaged over 5 runs. Here MEAN ABS., MAXIMUM, and MINIMUM represent the mean absolute fairness value, the fairness level of the most well-off group, and the fairness level of the worst-off group, respectively.

| METHOD (L) | ACCURACY ↑ | FAIRNESS | | | |
|---|---|---|---|---|---|
| | | MEASURE | MEAN ABS. ↓ | MAXIMUM | MINIMUM |
| Unconstrained | $0.7905 \pm 0.0033$ | AP | $0.0131 \pm 0.0021$ | $0.0123 \pm 0.0021$ | $-0.0138 \pm 0.0022$ |
| Constant | $0.666 \pm 0.0$ | AP | $0.053 \pm 0.0$ | $0.056 \pm 0.0$ | $0.051 \pm 0.0$ |
| Weighted ERM | $0.7906 \pm 0.0032$ | AP | $0.0127 \pm 0.0023$ | $0.0119 \pm 0.0022$ | $-0.0134 \pm 0.0024$ |
| Constrained | $0.467 \pm 0.115$ | AP | $0.036 \pm 0.003$ | $0.039 \pm 0.003$ | $0.034 \pm 0.003$ |
| FairGrad | $0.7837 \pm 0.0049$ | AP | $0.0023 \pm 0.0009$ | $0.0023 \pm 0.001$ | $-0.0022 \pm 0.0008$ |
| Unconstrained | $0.789 \pm 0.0026$ | Eodds | $0.0301 \pm 0.011$ | $0.0377 \pm 0.0153$ | $-0.0458 \pm 0.0184$ |
| Constant | $0.667 \pm 0.0$ | Eodds | $0.0 \pm 0.0$ | $0.0 \pm 0.0$ | $0.0 \pm 0.0$ |
| Weighted ERM | $0.7886 \pm 0.0032$ | Eodds | $0.0294 \pm 0.012$ | $0.0364 \pm 0.0169$ | $-0.0443 \pm 0.0206$ |
| Constrained | $0.663 \pm 0.032$ | Eodds | $0.008 \pm 0.003$ | $0.013 \pm 0.004$ | $0.004 \pm 0.002$ |
| BiFair | $0.768 \pm 0.007$ | Eodds | $0.008 \pm 0.005$ | $0.011 \pm 0.006$ | $-0.011 \pm 0.008$ |
| FairBatch | $0.788 \pm 0.0027$ | Eodds | $0.0045 \pm 0.0033$ | $0.0069 \pm 0.0065$ | $-0.0063 \pm 0.0049$ |
| FairGrad | $0.7885 \pm 0.0027$ | Eodds | $0.0043 \pm 0.0019$ | $0.0073 \pm 0.0037$ | $-0.0068 \pm 0.0045$ |
| Unconstrained | $0.7902 \pm 0.0038$ | Eopp | $0.0094 \pm 0.0031$ | $0.0162 \pm 0.0053$ | $-0.0215 \pm 0.0071$ |
| Constant | $0.667 \pm 0.0$ | Eopp | $0.0 \pm 0.0$ | $0.0 \pm 0.0$ | $0.0 \pm 0.0$ |
| Weighted ERM | $0.7893 \pm 0.0031$ | Eopp | $0.009 \pm 0.003$ | $0.0155 \pm 0.0051$ | $-0.0206 \pm 0.0069$ |
| Constrained | $0.706 \pm 0.002$ | Eopp | $0.004 \pm 0.0$ | $0.01 \pm 0.001$ | $0.0 \pm 0.0$ |
| BiFair | $0.77 \pm 0.002$ | Eopp | $0.019 \pm 0.01$ | $0.033 \pm 0.017$ | $-0.044 \pm 0.023$ |
| FairBatch | $0.79 \pm 0.0031$ | Eopp | $0.0012 \pm 0.0015$ | $0.0022 \pm 0.0026$ | $-0.0026 \pm 0.0034$ |
| FairGrad | $0.7893 \pm 0.0026$ | Eopp | $0.0011 \pm 0.0009$ | $0.0024 \pm 0.002$ | $-0.0021 \pm 0.0016$ |

Table 22: Results for the Folktables Adult dataset with Non Linear Models. All the results are averaged over 5 runs. Here MEAN ABS., MAXIMUM, and MINIMUM represent the mean absolute fairness value, the fairness level of the most well-off group, and the fairness level of the worst-off group, respectively.

| METHOD (NL) | ACCURACY ↑ | FAIRNESS | | | |
|---|---|---|---|---|---|
| | | MEASURE | MEAN ABS. ↓ | MAXIMUM | MINIMUM |
| Unconstrained | $0.8037 \pm 0.0037$ | AP | $0.0131 \pm 0.0017$ | $0.0123 \pm 0.0016$ | $-0.0139 \pm 0.0017$ |
| Constant | $0.666 \pm 0.0$ | AP | $0.053 \pm 0.0$ | $0.056 \pm 0.0$ | $0.051 \pm 0.0$ |
| Weighted ERM | $0.8046 \pm 0.0049$ | AP | $0.0131 \pm 0.0014$ | $0.0123 \pm 0.0014$ | $-0.0138 \pm 0.0015$ |
| Adversarial | $0.8016 \pm 0.0053$ | AP | $0.0122 \pm 0.0016$ | $0.0115 \pm 0.0015$ | $-0.0129 \pm 0.0016$ |
| FairGrad | $0.7917 \pm 0.0025$ | AP | $0.0016 \pm 0.0011$ | $0.0016 \pm 0.0011$ | $-0.0016 \pm 0.001$ |
| Unconstrained | $0.7947 \pm 0.0078$ | Eodds | $0.0314 \pm 0.0059$ | $0.0373 \pm 0.0058$ | $-0.0454 \pm 0.0066$ |
| Constant | $0.667 \pm 0.0$ | Eodds | $0.0 \pm 0.0$ | $0.0 \pm 0.0$ | $0.0 \pm 0.0$ |
| Weighted ERM | $0.7902 \pm 0.0049$ | Eodds | $0.0327 \pm 0.0061$ | $0.04 \pm 0.0067$ | $-0.0488 \pm 0.0077$ |
| Adversarial | $0.806 \pm 0.0047$ | Eodds | $0.0035 \pm 0.0018$ | $0.0051 \pm 0.0021$ | $-0.0053 \pm 0.0028$ |
| BiFair | $0.793 \pm 0.006$ | Eodds | $0.006 \pm 0.003$ | $0.007 \pm 0.003$ | $-0.007 \pm 0.004$ |
| FairBatch | $0.8061 \pm 0.0044$ | Eodds | $0.0051 \pm 0.0015$ | $0.0087 \pm 0.0048$ | $-0.0084 \pm 0.0029$ |
| FairGrad | $0.7997 \pm 0.0087$ | Eodds | $0.0045 \pm 0.0029$ | $0.0067 \pm 0.0045$ | $-0.0071 \pm 0.0058$ |
| Unconstrained | $0.7902 \pm 0.0044$ | Eopp | $0.0097 \pm 0.0026$ | $0.0168 \pm 0.0045$ | $-0.0222 \pm 0.006$ |
| Constant | $0.667 \pm 0.0$ | Eopp | $0.0 \pm 0.0$ | $0.0 \pm 0.0$ | $0.0 \pm 0.0$ |
| Weighted ERM | $0.7947 \pm 0.0022$ | Eopp | $0.0105 \pm 0.0027$ | $0.0181 \pm 0.0047$ | $-0.024 \pm 0.0062$ |
| Adversarial | $0.8108 \pm 0.0161$ | Eopp | $0.0034 \pm 0.0057$ | $0.0041 \pm 0.0057$ | $-0.0095 \pm 0.017$ |
| BiFair | $0.793 \pm 0.008$ | Eopp | $0.028 \pm 0.017$ | $0.048 \pm 0.029$ | $-0.064 \pm 0.039$ |
| FairBatch | $0.8038 \pm 0.0063$ | Eopp | $0.0008 \pm 0.0005$ | $0.0014 \pm 0.0009$ | $-0.0018 \pm 0.0012$ |
| FairGrad | $0.8058 \pm 0.0035$ | Eopp | $0.0014 \pm 0.0014$ | $0.003 \pm 0.0031$ | $-0.0026 \pm 0.0024$ |

