# OpenReview forum: "FairGrad: Fairness Aware Gradient Descent"
_ICLR.cc/2023/Conference — Submitted to ICLR 2023_

### Official Review · Reviewer_BapZ · 2022-10-26

**Confidence:** 3
**Correctness:** 3
**Technical Novelty And Significance:** 2
**Empirical Novelty And Significance:** 2
**Recommendation:** 5

**Clarity, Quality, Novelty And Reproducibility:**

For the details of these points, please see my review of "Strength and Weakness". However, to give a high-level answer:

**Clarity:** The overall idea is clear. The notation may sometimes get confusing.

**Quality:** The writing of the paper has a very good quality.

**Novelty:** I believe the paper has some novelty in the general fairness definition (S2.1), however, the main idea and the solution technique look very standard.

**Reproducibility:** The paper is reproducible because the steps used in the proposed algorithm are explained in detail and a Python package is provided.

**Strength And Weaknesses:**

**Strength:** I believe the literature is summarized very well, and the paper is written well in general. The motivation is clear and the topic covered is relevant to the research our community is conducting. I really like equation (1), and the related proofs in Example 1 and Appendix B.

**Weaknesses:**
*I list my major and minor concerns that made me recommend a rejection. However, I still like some aspects of the paper, hence I hope that my comments would be found useful by the authors to improve the paper's quality. I am also going to stay active during the discussion period in case the authors have updates or questions regarding my review.*

***Major Weaknesses/Questions***
- I think this framework is just providing us with a higher-level notation for the fair classifier optimization problem, but unless I am not seeing something, we already know these gradient descent updates in each of the covered settings. For example, take the algorithm explained in 3.1, and replace $F_k(\mathcal{T}, h_\theta)$ with the specific definition of Accuracy Parity (AP). Then the resulting gradient descent algorithm is already known. Same for any other fairness notion covered in this work. In general, I do not think there is much advantage that comes with the new expression $F_k(\mathcal{T}, h_\theta)$ in addition to providing some intuition.
- The optimization/duality/GD discussions are lacking assumptions, discussions, and citations. The convexity of the functions is never discussed, but duality is being used. Convergence to local solutions instead of global solutions is not 'warned'. The paper concentrates on the computational tools and numerical experiments that the theoretical arguments look weaker.
- The proposed 'weighting' is obtained immediately in the alternating GD setting. There is no selection of these weights, rather everything is a corollary of using GD.
- Lemma 1 looks redundant, as negative weights come automatically unless we rule them out.
- Further questions and minor weaknesses that are enumerated next.

***Minor Weaknesses/Questions***
- Introduction: Maybe the authors can mention that the designing phase of the algorithms has access to the so-called sensitive attributes, as there is a new line of research where the optimizer is oblivious. See, e.g., Mozannar, H., Ohannessian, M., & Srebro, N. ICML (2020)
- In the introduction, the 'reweighting scheme' is not clear. This is understood later.
- Page 2: "They are also often limited in the range of ..." the word 'often' sounds a little subjective. Could the authors please consider adding citations to or deleting such sentences?
- Page 2: "gradient descent based methods" -> do the authors mean gradient descent based solutions instead?
- Page 2: "update the weights of the examples" is not clear yet.
- python -> I think 'p' should be capitalized
- Page 2: while discussing that the library will be very similar to PyTorch functions, could the authors please also discuss that the user needs to provide the sensitive attributes?
- I personally think the notation abuse of $\mathbb{P}(E)$ referring to both empirical and true probabilities is more than a slight notational abuse. For example, in equation (1), the term $\mathbb{P}(h_\theta(x) \neq y | \mathcal{T}_{k'})$ is hard to grasp.
- Section 2.1: When the authors state $F_k$ as a representation of whether or not a group is advantaged, maybe they can highlight this with respect to a classifier. Sometimes in fairness literature "advantaged" groups mean those who have historically been associated with better/higher rewards.
- Domain of $F_k$ includes the set of distributions, but $F_k(\mathcal{T}, h_\theta)$ takes a dataset. Probably the notation should be formalized with $\mathcal{T}$ denoting the **empirical distribution** of the training set instead.
- Maybe the authors can define $F_{(r)}$ notation uses $(r)$ as the group number $r$ refers to?
- "$0.01$ .. is a good rule of thumb" -> why? This sounds subjective.
- Section 3.4, "$80\%$ rule in the US" -> similarly, there are no references here.
-  Minor comment: In the appendix, the proof is re-sated as Lemma 2, maybe this can be fixed to Lemma 1 again?
- The notation $[1,K]$ is confusing as $k \in [1,K]$ is not continuous, hence we probably need the $k \in [K]$ shorthand instead.
- In the optimization problems `arg min` and `st.` are not aligned.
- The gradient on top of page 4 is missing its function.

***Optimization Related Issues/Questions***
- Firstly, problem (2) is very general and there are lots of very strong techniques proposed to solve special cases of this problem. I think the use of GDs should be motivated further.
- Shouldn't the Lagrangian dual problem flip the max and min operators (e.g., in (3))? Because the Lagrangian dual function first minimizes the Lagrangian function over the primal optimization parameters, and the Lagrangian dual problem maximizes the Lagrangian function over the dual variables $\lambda$?
- To use duality we need a lot of assumptions. Here nothing is mentioned. The compactness of the feasible sets, convexity, whether or not strong duality holds, whether the proposed algorithm converges to a global solution or not and whether it means the fairness is not satisfied due to the relaxations, are not discussed. **Update: The authors replied to this concern, hence I am marginally increasing my score.**
- The optimization problems use $\arg\min$ instead of $\min$. However, $\arg\min$ returns sets and the duality arguments are for optimal values instead. It is not very usual (even correct) to use $\arg\min$ in my view. Unless we try to address a specific optimal solution that belongs to the $\arg\min$ set, if we are referring to an optimization problem, the standard approach is to use $\inf$ (or $\min$ if existence is discussed).
- Does the use of surrogates result in the promised fairness level not holding anymore? In general, the proposed method might give *a* solution, but is it feasible? Overall, I am not very convinced with the current discussions.
- The values $\omega_k$ seem to be coming immediately from the gradients, am I wrong? I think they are an immediate result of splitting the updates of the primal and dual variables.
- When it comes to the proposed decomposition of alternatingly updating the primal and dual variables, there are many existing works already, hence I believe this approach is not new and looks like a less sophisticated version of the primal-dual methods that the literature uses.
- Does "Algorithm 1" add anything to the paper? Maybe it should be clear from Section 3.1.

**Summary Of The Paper:**

In this work, the authors concentrate on variants of the fair classification problem. They unify some of the most popular fairness notions in a general representation framework by defining a quantity whose magnitude reveals (dis-)advantage to specific groups. By using the aforementioned quantity, they represent the fair classification optimization problem, derive a gradient descent (GD) algorithm to solve this problem and provide intuition on the "weights" that are used to update the GD iterations by bridging them with the protected groups.

**Summary Of The Review:**

In general, the paper covers an interesting topic, provides a useful Python package, and has extensive numerical experiments; however, I think the work lacks novelty as the problems solved and the solution techniques are all known.

---

> ### Author Response · Authors · 2022-11-14
> **Discussion**
>
> We thank the reviewer for the helpful and detailed comments. It seems that our way of presenting the results was not properly conveying the key contributions of the paper. Below we propose an alternate way to present the main results that should more meaningfully show the interest of the approach.
>
> Consider the following reweighting problem:
>
> $\text{argmin}\_{h\_\theta \in \mathcal{H}} \sum\_{k=1}^K w\_k L\_k\left(h\_\theta\right)$
>
> where $w\_k$ are weights and $L\_k\left(h\_\theta\right)$ are group-wise losses. The underlying idea in re-weighting approaches in fairness is to choose weights that will lead to fair models. One can use static weights (Calders et al., 2009) or weights that evolve through the optimization process (Roh et al., 2020). Our approach can be seen as an instance of the latter where we choose the weights after $t$ iterations in a specific way, that is $w^{(t)}\_k = \left[\mathbb{P}(\mathcal{T}\_k) + \sum\_{k'=1}^K C\_{k'}^k \eta\_\lambda\sum\_{i=1}^t F\_{k'}(\mathcal{T},h\_\theta^{(i)})\right]$ where $\eta\_\lambda$ is a parameter controlling the speed at which fairness should be taken into account. In the paper, we show that this choice is not random and that it, in fact, comes from a connection with the standard formulations (Equation (2) in Section 3.1) used in fair machine learning methods based on constrained optimization. Interestingly, note that Agarwal et al. (2018) showed that constrained problems in fairness can be solved by a sequence of cost-sensitive problems that, in turn, can also be thought of as a re-weighting approach with particular positive weights (See Section 3.1 in their paper). However, they only do so for binary classification (while we also consider multi-class problems) and they do not exploit the connection to re-weighting approaches in the way we do (they focus on the cost-sensitive formulation).
>
> Having this in mind, we would like to clarify a few points that the reviewer raised.
>
> **Novelty of the method:** We agree with the reviewer that some parts of the paper are not novel:
>  - Equation (2) in Section 3.1 is not new and has been used in the fairness literature based on constrained optimization before (Agarwal et al., 2018, Cotter et al., 2019).
>  - Using Lagrange multipliers to solve constrained optimization is not new. In fact this is a completely natural approach that was used before in the fairness literature (Agarwal et al., 2018, Cotter et al., 2019).
>
> However, to the best of our knowledge, the following parts are new and valuable:
>  - There is an explicit connection between constrained optimization (Equation (2)) and re-weighting (Algorithm 1) with weights updated in a way that was not used before.
>  - An explicit consideration of the importance of negative weights (this is also related to the reviewer comment on the redundancy of Lemma 1). It is worth noting that in previous re-weighting approaches, a frequent assumptions is that the weights should be positive (Roh et al., 2020, Iosifidis and Ntoutsi, 2019, Jiang and Nachum, 2020) but here we show that having negative weights is necessary in some cases. It makes the problem harder to solve as we will show in our answer to the comment on the lack of discussions of the theoretical aspects.
>
> **Interest of $F\_k(\mathcal{T}, h\_\theta)$:** In the fair machine learning community it is known that several group fairness definitions have similar forms (equalized odds, equality of opportunity) and a method proposed for one is applicable to the other. Here, by giving a proper and general definition of $F\_k(\mathcal{T}, h\_\theta)$, we explicitly delimit the applicability of FairGrad and provide a way to reason about several fairness notions simultaneously.
>
> **Lack of discussions of the theoretical aspects:** We initially did not discuss these points as, unfortunately, FairGrad does not come with such guarantees. More precisely, even when we use a convex loss, the overall problem might remain non-convex because of the potentially negative weights. In other words, we do not have formal convergence guarantees here. In fact, before these considerations, we do not even have guarantees that a proper set of weights that would lead to a solution exists and that duality is applicable here. However, in practice, FairGrad behaves well and seems to converge to meaningful solutions as can be seen in the experiments. We agree that this is indeed something that should be explicitly mentioned, thus we added a footnote about this issue in Section 3.1 in the main paper.
>
> **Everything is a corollary of using GD:** We hope that the previous discussion on re-weighting and constrained optimization helped clarify this point. To summarize, we agree that the weights come quite straightforwardly from our use of GD. However, we think that this is not necessarily a weakness but rather an interesting outlook on the problem of fair machine learning as it connects re-weighting approaches and constrained optimization approaches.

---

> > ### Comment · Reviewer_BapZ · 2022-11-16
> > **Acknowledging the answer & further question**
> >
> > Dear Authors,
> >
> > Thank you very much for your reply! And apologies for my slightly delayed reply.
> >
> > I read your responses to other reviewers, mostly clear, but I will be interested in joining the discussions with Reviewers 93K5 and chVG if possible.
> >
> > Regarding my review response: The discussion about reweighting is clear! I would like to also thank you for your honest but detailed answer on convexity/convergence.
> >
> > I would like to kindly ask you what the novelty of this work is from your view. You can be very direct in your answer, and no need to be humble. For me, this is still not very clear. In general, if we already know all these fairness notions, have algorithms for each notion, and also have papers out there showing weighting settings, what is the punchline of this method? Given that we do not necessarily have convergence guarantees or any novel analysis of this algorithm, I am curious about the potential benefits of using this work.
> >
> > Moreover, apologies if you answered this somewhere, but it is still not clear to me whether the relaxations or local convergence in this paper would result in the fairness constraints being violated. Dual variables will be associated with these concerns, and if the dual variables do not converge to a global solution, or if the relaxations taken in this paper lead to a loss of fairness, then this would be beneficial to discuss.
> >
> > Many thanks for your time.

---

> > > ### Author Response · Authors · 2022-11-16
> > > **Discussion**
> > >
> > > Dear Reviewer BapZ,
> > >
> > > Thank you for the time you spent considering this paper. Please find below our answers to your new questions. Do not hesitate if you require additional details.
> > >
> > > **Novelties of this work:**
> > > First, we noticed that a line skip was missing in the novelty discussion in our previous answer. It made it look like we only mentioned the parts that were not novel while we also mentioned some novelties. We edited our answer to add this line skip back and make it clearer. We summarize the novelties below:
> > >  - A new reweighting approach to learn fair models in machine learning that is empirically competitive across a wide range of problems. Indeed, while other reweighting approaches exist, they use different sets of weights.
> > >  - The way we leverage the connection between reweighting methods and constrained optimization approaches in fairness to obtain meaningful weights. Indeed, this connection was mentioned in the literature (Agarwal et al., 2018) but was not exploited in the way we do.
> > >  - A Python implementation of the method that is simple to use and has minimal overhead.
> > >  - An explicit statement that using negative weights is sometimes necessary in re-weighting methods to achieve exact fairness.
> > >
> > > **Impact of relaxation and local convergence on fairness:**
> > > To reason about this question, let us assume that the approach converged, that is both the dual variables ($\lambda$) and the primal variables ($\theta$) converged to a solution. Then, it necessarily means that this solution is fair otherwise the dual variables would change given our update rule. The relaxation does not impact this reasoning since it does not affect the update rule of the dual variables.
> > >
> > > Note that, in case the primal variables converge but not the dual ones, that is the fairness level of the solution is such that the successive weights all lead to zero vectors gradients for the primal variables, then we cannot guarantee that the model is fair. In this case, the relaxation has an impact since it affects the gradients of the primal variables.

---

> > > > ### Comment · Reviewer_BapZ · 2022-11-17
> > > > **Discussion Follow-up**
> > > >
> > > > Dear Authors,
> > > >
> > > > Thank you very much for your reply. The discussion on the implications of local convergence on fairness is mostly clear. When the algorithm terminates then fairness is satisfied necessarily -- in that case, do you think this is worthy to mention (formally with appropriate terminology) in the text?
> > > >
> > > > I understand your comments on the novelty, thank you very much for your time. I believe the connection between equation (2) and reweighting is not very surprising, given that reweighting is an immediate consequence of the alternating descent method and this is already a method to solve (2).
> > > >
> > > > I will increase my score marginally (mainly because of the fact that fairness will not be violated in a 'hidden' step), and follow the discussions that will follow carefully. Please let me know if you would like to discuss anything further.

---

### Official Review · Reviewer_chVG · 2022-10-26

**Confidence:** 4
**Correctness:** 3
**Technical Novelty And Significance:** 3
**Empirical Novelty And Significance:** 3
**Recommendation:** 6

**Clarity, Quality, Novelty And Reproducibility:**

The paper is quite readable and relatively clear. The Fairgrad approach builds on the previously proposed approaches related to Lagrangian formulations of fair classification.

**Strength And Weaknesses:**

### Strength

In my opinion, the primary strength of this work is the python package that is being released as part of this paper. Given the package, the proposed formulation can then be applied across a variety of different settings more easily. In my opinion, this alone, is an important enough contribution of this paper.


### Weaknesses

- Novelty in formulation: As a reviewer, I actually don't place any emphasis on the 'novelty' of a work, so I don't see this as a weakness. My only point here is that the formulation here is not different that those in [Jiang & Nachum, Cotter et al.] and others that all employ the Langrange multplier formulation. The key difference seems to be that this formulation allows for negative weights/multipliers which other formulations don't. I think the paper is written to introduce fairgrad as a new method, but in my opinion it is actually a generalization of existing methods. Perhaps a better pitch for this paper would be that these existing methods are generalized and implemented as part of clean python/pytorch package. In my opinion, that alone, is a worthy ICLR submission. The push to make Fairgrad a unique method  seems a stretch to me. I am willing to change my opinion if the authors believe that this is not the case.

- The empirical results are helpful, but I would recommend that the authors jettison the traditional Adult dataset and use the newly proposed Adult dataset from the Neurips 2021 paper by Ding et. al. titled retiring adult. Results on that dataset would be more convincing.

- In the experiments, reporting the average fairness level for all groups is slightly misleading since all the improvement might be in the dominant group. I think these should be broken into groups.



**Summary Of The Paper:**

This paper presents a python package and an approach called fairgrad. The goal of fairgrad is to learn model parameters that satisfy accuracy parity or its approximate equivalent across groups of a dataset. Fairgrad does this by formulating a constrained optimization problem, which is to minimize a model perfomance metric subject to accuracy parity constraints across demographic groups that partition the training dataset. This formulation can then be relaxed and put in a langrangian form where the lagrange multipliers are applied to the accuracy parity constraint for each group. This entire formulation can then be 'solved' via an alternating gradient descent formulation. The first component of the alternating gradient descent formulation applied gradient descent to the langrange multipliers. Given the current setting of the langrange multipliers, gradient descent is then applied again to the model parameters. This scheme is iterated until convergence or a pre-specified number of iterations. The Fairgrad package implements this scheme automatically. The empirical performance of the proposed scheme is demonstrated on a variety of benchmarks.

**Summary Of The Review:**

The paper proposed a formulation, FairGrad, that is presented in a python package that is then tested across a wide variety of datasets. Overall, the proposed approach can be easily applied by others in the community.

---

> ### Author Response · Authors · 2022-11-14
> **Discussion**
>
> We would like to thank the reviewer for the feedback and the positive view on the paper. We try to clarify our stance in terms of novelty of the paper below and we briefly comment additional experiments on Folktables (Ding et al., 2021) that were added to the paper at the end of Section E.3 of the appendix.
>
> **Novelty in the formulation:** To answer this concern we would like to more explicitly summarize which part of our approach are novel and which part are not. Let us start with the parts that already exist in the literature:
> - Equation (2) in Section 3.1 is not new and has been used in the fairness literature based on constrained optimization before (Agarwal et al., 2018, Cotter et al., 2019).
>  - Using Lagrange multipliers to solve constrained optimization is not new. In fact this is a completely natural approach that was used before in the fairness literature (Agarwal et al., 2018, Cotter et al., 2019).
> However, to the best of our knowledge, the following parts are new:
>  - There is an explicit connection between constrained optimization (Equation (2), Section 3.1) and re-weighting (Algorithm 1) with weights updated in a way that was not used before. To give the full picture here it is worth noting that Agarwal et al. (2018) showed that constrained problems in fairness can be solved by a sequence of cost-sensitive problems that, in turn, can also be thought of as a re-weighting approach with particular positive weights (See Section 3.1 in their paper). However, they only do so for binary classification (while we also consider multi-class problems) and they do not exploit the connection to re-weighting approaches in the way we do (they focus on the cost-sensitive formulation).
>  - An explicit consideration of the importance of negative weights. It is worth noting that in previous re-weighting approaches, a frequent assumptions is that the weights should be positive (Roh et al., 2020, Iosifidis and Ntoutsi, 2019, Jiang and Nachum, 2020) while here we show that having negative weights is necessary.
>
> **Experiments on Folktables (Ding et al., 2021):**
> We used the California census data with gender as the sensitive attribute. There are $195665$ instances, with 9 features describing several attributes of a person. We use the same pre-processing step as recommended by the authors. Over this dataset, we notice that FairGrad and FairBatch have very similar performances and generally outperform the other approaches.
>
> **Groupwise fairness level:** We agree that the average fairness level does not tell the full extent of the story and that considering each group independently is important. This is why we report the fairness level of the worst-off group (Minimum) and the most well-off group (Maximum) in the supplementary.

---

> > ### Comment · Reviewer_chVG · 2022-11-18
> > **Thanks for the response**
> >
> > I have read the author response and those for the other reviewers. I believe the authors have answered most of the questions I have about the paper. It looks like one of the key contributions here is the use of negative weights, and I believe the justification here mostly makes sense. I actually do believe the code and package here will also be quite useful to the community.
> >
> > Thanks

---

### Official Review · Reviewer_93K5 · 2022-10-27

**Confidence:** 4
**Clarity, Quality, Novelty And Reproducibility:** see my comments above.
**Correctness:** 3
**Technical Novelty And Significance:** 2
**Empirical Novelty And Significance:** 2
**Recommendation:** 3

**Strength And Weaknesses:**

+:

The performance of the algorithm seems to be better than the state of the arts.

The paper writing is clear.

-:

The technical novelty seems limited. The difference between this work and the previous work [Agarwal et al., 2018] seems very marginal. The only difference here is that this work uses a gradient descent ascent algorithm while Agarwal et al. used the online convex optimization algorithm (exponentiated gradient) with the best response for one of the two players.

Furthermore, the difference between this work and the previous work [Roh et al., 2020] also seems marginal -- (1) min max optimization vs bilevel optimization and (2) negative group weights are allowed.

The breadth of baseline algorithms is somewhat limited. Many new open-source frameworks implement a large number of fairness algorithms, and I believe that they will be helpful in running more extensive evaluations.

Also, many of the algorithms have tuning knobs, so it's more helpful if one compares the entire trade-offs achieved by each method.

**Summary Of The Paper:**

The authors propose group reweighting algorithms for achieving group fairness.

**Summary Of The Review:**

see my comments above.

---

> ### Author Response · Authors · 2022-11-14
> **Discussion**
>
> We would like to thank the reviewer for their feedback. We clarify the differences between FairGrad and existing approaches below.
>
> **Novelty compared to other methods:**
>  - Agarwal et al., 2018: This paper indeed starts with a formulation similar to Equation (2) in our paper and also uses Lagrange multipliers (Equation (3)). We also agree that they show that constrained problems in fairness can be solved by a sequence of cost-sensitive problems that, in turn, can also be thought of as a re-weighting approach with particular positive weights (See Section 3.1 in their paper and note that we allow negative weights). However, we believe that the similarities end here and that there is fundamental differences between our approaches. First, they do not exploit the connection to re-weighting approaches in the way we do since they focus on the cost-sensitive formulation. Second, they only consider binary classification while FairGrad can handle multi-class problems. Third, their goal is different from ours since they aim to learn a distribution over the hypotheses while we try to find the best hypothesis in the class. Fourth, at each iteration of their algorithm they fully solve the cost-sensitive problem they obtain while we only take a single re-weighted gradient step.
>  - Roh et al., 2020: We thank the reviewer for pointing out these differences. We think that they are not so marginal since min max and bi-level optimization are two connected but different research fields and since we show that negative weights are sometimes necessary (Lemma 1).
>
> **Breadth of baselines:** Following the advice of Reviewer xqvP, we added another baseline which is a reweighting approach with weights inversely proportional to the size of the sensitive groups. Overall, it means that we now cover a fairly large spectrum of techniques:
>  - Bi-level optimization: Ozdayi et al., 2021 and Roh et al., 2020
>  - Re-weighting approaches: baseline suggested by Reviewer xqvP and Roh et al., 2020
>  - Constrained Optimization: Cotter et al. 2018
>  - Adversarial Learning: Adversarial method based on Raff, E. and Sylvester, J., 2018
>
> Furthermore, the main conclusion of our experiments is that there is no clear winner among all the compared methods (including FairGrad) and all of them have settings where they perform better than the others.

---

### Official Review · Reviewer_xqvP · 2022-11-03

**Confidence:** 4
**Correctness:** 4
**Technical Novelty And Significance:** 3
**Empirical Novelty And Significance:** 3
**Recommendation:** 6

**Clarity, Quality, Novelty And Reproducibility:**

Clarity

+ The paper is clearly written and the concepts are explained well. The experiments are clear and the results are fairly represented for the most part.

Quality

+ The submission is of high quality. Barring some changes to the experiments, overall the paper is high-quality

Novelty

+ the paper considers an important problem and proposes a novel algorithmic solution to the problem.

Reproducibility

+ the paper presents the psuedo code and for the most part the experiments seem reproduccible. The paper also links to the code.

**Strength And Weaknesses:**

Strength

+ Although the paper positions this as a problem of addressing fairness, this method can be applied more broadly to any domain where the dataset contains multiple distributions, and we would like the hypothesis to not reflect the same underlying distribution from which the dataset is generated and learn to generalize across the various classes equally well. Fairness is one such example, but there are many practical applications to this such as data coming from different distributions (e.g., dataset collected over many days from a slowly changing environment).

+ The procedure is very simple to implement, and can be readily accommodated in a typical SGD training loop. Moreover, the additional overhead is very minimal since the aditional information the algorithm requires (i.e., average accuracy per group) is anyway computed in a typical SGD update step.

Weakness

+ The first weakness of the paper I find is that the paper does not compare to more baselines that are natural. First, since we know the groups already, one could in-principle first compute the ratios of prevalence of the various group in a random sample of the dataset and use an IPS estimate to correct for this bias (i.e., weight_i = 1/Pr[random sample belongs to class i]). The advantage with this method is that, this is direct and one does not need to worry about learning additional parameters. Second, one could try to learn the weights lambda outside of the SGD update using some form of non-differentiable method (such as BO) which are much more sample efficient.

+ The second weakness of this paper is that the paper does not try this method on more modern datasets/architectures (e.g., GLUE/superGLUE and transformer). This may be needed, because even minor additional overhead during training will be more visible when training on large models on big datasets. I would recommend the authors to perform ablations on overhead more thoroughly and/or add the caveat that the overhead could be minimal but more experiments are needed.


--- EDIT

The authors have run additional experiments to resolve both my main worries: scalability as a function of architecture complexity and natural baselines. So I am increasing my score.

**Summary Of The Paper:**

In this paper, the authors propose a modification to Stochastic gradient descent where they alternatively learn the lagrange multipliers for the fairness constraint along with learning the parameters of the model during training. Fairness here is modelled as a constraint, and different data-points are associated with different weights (one weight per class). They empirically show the power of this method, and analyze the accuracy/fairness tradeoff obtained by this method and compare it against full constrained optimization.

**Summary Of The Review:**

Overall, the paper is well-written and the direction is important and promising. However, I would like the authors to address the above weakness in a meaningful form so that the submission can be made even more stronger. given a reasonable response to the above, I would be able to move my score based on that.

---

> ### Author Response · Authors · 2022-11-14
> **Discussion**
>
> We would like to thank the reviewer for the helpful comments that will help us improve the experiments in the paper. We added some of the experiments suggested by the reviewer to the paper, we briefly comment and summarize them below.
>
> **Considering more natural baselines:** We used the weight suggested by the reviewer to learn models on all our datasets, that is the loss of each example is multiplied by the inverse of the probability of belonging to this example's sensitive group. To summarize the results, the performance of this method is not noticeably better than that of the Unconstrained method.  This is not so surprising as the size of a sensitive subgroup is not necessarily correlated with the level of unfairness of the model, in particular even balanced datasets might lead to unfair models.
>
> Regarding the BO baseline, we are not sure we understand the idea exactly. Could you please refer us to a fairness paper implementing the method you are referring to?
>
> **Training overhead:** To evaluate the overhead of our method, we report the wall clock time, in seconds, to train for an epoch with the Unconstrained approach and our method over various datasets, models, and settings. More specifically,
>  - We show the effect of model size by varying the number of hidden layers of the model over the Adult Income dataset, which consists of $45,222$ records. We used an Intel Xeon E5-2680 CPU to train over this dataset.
>  - A large transformer model (bert-base-uncased) fine tuned over the Twitter Sentiment Dataset consisting of $200k$ English tweets. We trained the model using a Tesla P100 GPU.
>  - A large convolutional neural network-based model (resnet18) fine tuned over the UTK-Face dataset consisting of $23,708$ images. We trained the model using a Tesla P100 GPU.
>
> | Setting                                            | Model Parameters | Batch Size | Unconstrained Time | Fairgrad        | Delta |
> |----------------------------------------------------|------------------|------------|--------------------|-----------------|-------|
> | Linear model  - Adult Dataset -CPU                 | 106              | 512        | 0.277 ± 0.031      | 0.307 ± 0.01    | 0.03  |
> | Non linear model with 2 layer  -Adult Dataset -CPU | 1762             | 512        | 0.315 ± 0.036      | 0.316 ± 0.029   | 0.01  |
> | Non linear model with 5 layer  -Adult Dataset -CPU | 21346            | 512        | 0.370 ± 0.042      | 0.394 ± 0.025   | 0.02  |
> | Non linear model with 10 layer -Adult Dataset -CPU | 39042            | 512        | 0.483 ± 0.021      | 0.499 ± 0.034   | 0.02  |
> | Non linear model with 20 layer -Adult Dataset -CPU | 80642            | 512        | 0.672 ± 0.034      | 0.689 ± 0.026   | 0.02  |
> | Resnet18 trained -UTKFace -GPU                     | 11177538         | 64         | 31.173 ± 0.085     | 31.588 ± 0.055  | 0.42  |
> | Bert (Transformer)  Twitter Sentiment -GPU         | 109505310        | 32         | 2246.342 ± 3.20    | 2294.382 ± 4.01 | 48.04 |
>
> The overhead we observe is limited and should not be critical in most applications as it does not depend on the complexity of the model but, instead, on the number of examples and the batch size. This is not surprising as FairGrad has two overheads:
>  - Initial computations of the group proportions and the constants $C$. This is only done once at the start of the optimization procedure since the values can be stored. It only depends on the number of samples used to approximate the values and the number of groups. In particular, it does not depend on the model.
>  - In each epoch we take take several gradient steps with a given batch size. In each gradient step, we need to compute the fairness level of the model on the current batch and to compute the weighted loss. The predictions of the model for the batch are already available since they are necessary to compute the gradient and there is no overhead in the calculation of these predictions since our approach has no computational overhead during inference. Furthermore, computing the gradient of a weighted loss is as expensive as computing the gradient of a standard loss (which can be seen as a loss with uniform weights over the examples). Overall, it means that the overhead in each gradient step should be small compared to the cost of computing the gradient as it does not depend on the number of parameters in the model.

---

### Decision · Program_Chairs · 2023-01-20

**Decision:**

Reject

**Justification For Why Not Higher Score:**

The apparent somewhat-less novelty is a barrier to acceptance in the paper's current form.

**Justification For Why Not Lower Score:**

N/A

**Metareview: Summary, Strengths And Weaknesses:**

This work develops a modification to Stochastic Gradient Descent that---via an alternating (descent) approach--- learns the Lagrange multipliers for fairness (which is modeled as a constraint) along with learning the parameters of the model during training. Experimental results show the utility of the approach; the paper also analyzes the algorithm’s accuracy--fairness tradeoff as compared to full constrained optimization. The method is easily implementable and the paper is well-written.

One of the primary negatives is that the novelty factor, as seen by the authors, should be brought out much better by them. As an example, the connection between eq. (2) and the reweighting is not very surprising, given that reweighting is an immediate consequence of alternating descent and this is already a method to solve (2). This is not to discourage the authors, but to ask them to think carefully through the merits and novelty of their method.